



# Joint observation-model mixed-layer heat and salt budgets in the eastern tropical Atlantic

Roy Dorgeless Ngakala[1,2], Gael Alory[3], Casimir Yélognissè Da-Allada[1,4,5], Olivia Estelle Kom[1], Julien Jouanno[3], Willi Rath[6], and Ezinvi Baloïtcha[2]

[1]Department of Oceanography and Applications, International Chair in Mathematical Physics and Applications, University of Abomey-Calavi, Cotonou, Benin,
[2]Department of Oceanography and Environment, Institut National de Recherche en Sciences Exactes et Naturelles, Pointe-Noire, Congo,
[3]Laboratoire d'Etudes en Géophysique et Océanographie Spatiales, University of Toulouse, Toulouse, France,
[4]Laboratoire de Géosciences, de l'Environnement et Applications, Université Nationale des Sciences Technologies, Ingénierie et Mathématiques, Abomey, Benin,
[5]Laboratoire d'Hydrologie Marine et Côtière, Institut de Recherches Halieutiques et Océanologiques du Bénin, Cotonou, Benin,
[6]GEOMAR Helmholtz-Zentrum für Ozeanforschung Kiel, Germany.

*Correspondence to*: Roy Dorgeless Ngakala (roy.ngakala@gmail.com)

**Abstract.** In this study, we use a joint observation-model approach to investigate the mixed-layer heat and salt annual mean and seasonal budgets in the eastern tropical Atlantic. The regional PREFCLIM observational climatology provides the budget terms with a relatively low spatial and temporal resolution compared to the online NEMO model, and this later is then resampled as in PREFCLIM climatology. In addition, advection terms are recomputed offline from the model as PREFCLIM gridded advection computation. In Senegal, Angola and Benguela regions, the seasonal cycle of mixed-layer temperature is mainly governed by surface heat fluxes; however, it is essentially driven by vertical heat diffusion in Equatorial region. The seasonal cycle of mixed-layer salinity is largely controlled by freshwater flux in Senegal and Benguela regions; however, it follows the variability of zonal and meridional salt advection in Equatorial and Angola regions respectively. Our results show that the time-averaged spatial distribution of NEMO offline heat/salt advection terms compares much better to PREFCLIM horizontal advection terms than the online heat/salt advection terms. However, the seasonal cycle of horizontal advection in selected regions shows that NEMO offline terms do not always compare well with PREFCLIM, sometimes less than online terms. Despite this difference, these results suggest the important role of small scale variability in mixed-layer heat and salt budgets.

## 1 Introduction

The interaction between ocean and atmosphere plays a crucial role in the climate system. This interaction involves heat and freshwater fluxes, which affect temperature and salinity variations in the upper oceanic mixed-layer. Therefore, sea surface



temperature and salinity (SST, SSS) are two key climate variables and understanding the balance of processes determining their variability is a key requirement to accurately simulating the climate system.

In the Eastern tropical Atlantic, seasonal climate variability is mostly associated with the West African Monsoon (WAM),
and strongly linked with SST and SSS variations. SST is here characterized by a strong seasonal cycle, which influences the regional climate, particularly the large scale atmospheric circulation and rainfall over the ocean and African continent (Carton and Zhou, 1997; Foltz et al., 2013; Kushnir et al., 2006). In the Gulf of Guinea, the seasonal formation of the Atlantic cold tongue (ACT), associated with equatorial upwelling, is the main feature associated with SST seasonal variations (Chang et al., 2006; Peter et al., 2006; Giordani and Caniaux, 2001). This seasonal cooling creates an intense meridional SST front that
enhances Southern trade winds, shifts the Intertropical convergence zone (ITCZ) northward, which triggers the WAM (Philander and Pacanowski, 1981; Picaut, 1983; Waliser and Gautier, 1993; Caniaux et al., 2011). In the Angola-Benguela frontal zone off south-western Africa, SST variability has also an impact on coastal precipitation (Reason and Rouault, 2006). Conversely, the atmospheric conditions in the Eastern tropical Atlantic impact SST, as trade winds are the main driver of Eastern Boundary Upwelling Systems found along the coasts of Angola-Namibia and Senegal-Mauritania.

Like SST, SSS is an essential climate variable. Its variations are closely linked to the global hydrological cycle, as freshwater exchanges between ocean and atmosphere control its mean large-scale distribution (Durack and Wijffels, 2010; Bingham et al., 2012): regions of high SSS are dominated by evaporation while regions of low SSS are dominated by precipitation. In the Eastern tropical Atlantic, the intense precipitation under the ITCZ has a strong impact on seasonal variability of SSS. For example, off Senegal and Guinea, SSS decreases in boreal fall following the period when ITCZ reaches its most northern
position, which leads to a maximum in precipitation and runoff from Senegal and Gambia rivers (Camara et al., 2015). The Eastern tropical Atlantic is indeed characterized by low SSS plumes due to strong river discharges, including the Congo River (the second largest outflow in the world after the Amazon River) and the Niger River in the Gulf of Guinea. SSS variations are also associated with upwelling. In the equatorial Atlantic, SSS increases during the development of the Atlantic Cold Tongue (ACT) in boreal spring/summer (Schlundt et al., 2014; Da-Allada et al., 2014). Summer upwelling at the northern
coast of the Gulf of Guinea also increases coastal SSS (Alory et al., 2021). Off Angola, relatively high SSS is observed during the upwelling season in August while low SSS appears in March and October/November (Kopte et al., 2017; Awo et al., 2022). In the Benguela upwelling system further south, the maximum in SSS appears in April and the minimum in October (Junker et al., 2017).

SST and SSS have already been the focus of several studies in the Eastern tropical Atlantic. Studies only based on observations (in situ or/and satellite), and others combining observational data and model data, show a variety of physical processes contributing to heat and salt budget with a different balance from one region to another (Foltz et al., 2003; Foltz and McPhaden, 2008; Da-Allada et al., 2013, 2014).

In the northern tropical Atlantic far from the coast, the heat budget is largely driven by surface heat fluxes, essentially solar
radiations varying with the cloudy ITCZ position (Carton and Zhou, 1997). Off Northwest Africa in the Senegal region, the



seasonal cycle of SST is associated with coastal upwelling modulated by the seasonal variations of alongshore winds. In the equatorial zone, the mixed-layer heat budget is mainly controlled by surface heat fluxes, of which mainly the solar radiation is important (Carton and Zhou, 1997; Foltz et al., 2003; Yu et al., 2006). However, other oceanic processes can also play an important role in the seasonal heat budget. There is, e.g., zonal advection and vertical mixing which are important during the

formation of the ACT (Foltz et al., 2003; Hummels et al., 2014; Schlundt et al., 2014). However, the influence of vertical mixing in the heat budget remains low in the eastern (from 0°E to the coast of Africa) compared to the central equatorial Atlantic (Jouanno et al., 2011; Hummels et al., 2013). This is explained by the strong stratification in the eastern equatorial Atlantic, due to the transport of low-salinity and warm waters from the Gulf of Guinea, which contributes to reduce the vertical mixing. The low salinity is associated with the intense precipitation, and important freshwater intakes from the Niger and

Congo rivers in the Gulf of Guinea. Off Angola, annual variability of SST is driven by meridional wind stress, causing coastal upwelling, and thermocline depth variations forced by remote equatorial effects (Carton and Zhou, 1997).

In the northeast tropical Atlantic, including in the Senegal region, the salt budget is controlled by freshwater fluxes. The net mixed-layer salinity variations are, however, weak because of the compensation between the atmospheric and oceanic terms (Camara et al., 2015). In the eastern equatorial Atlantic and Gulf of Guinea, horizontal advection and vertical mixing play a

dominant role in determining the seasonal cycle of salt budget, which cannot be explained by freshwater fluxes only (Tzortzi et al., 2013; Da-Allada et al., 2013, 2014). This dominance of zonal advection and vertical processes extends southward and in the Angola coastal region (Camara et al., 2015).

In addition, it can be noted that various approaches have been used to estimate the heat and salt budgets in the tropical Atlantic,

and in particular the advection terms. Foltz et al. (2003) analyzed the mixed-layer heat balance at PIRATA mooring locations, where they computed the heat advection from monthly gridded climatologies of near-surface horizontal velocity, based on ship drifts and Lagrangian drifters, and SST gradient fields based on a combination of ship, buoy and satellite data. Wade et al. (2011) used a similar SST product but satellite-derived currents to estimate every 10 days the heat advection at positions of Argo profiles. Then monthly averages in 9 boxes covering the Gulf of Guinea were used to study the seasonal cycle of mixed-

layer heat as observed by Argo. Da-Allada et al. (2013) developed an original mixed-layer model of the tropical Atlantic at monthly, 1° resolution, where salinity was driven by observation-based climatological freshwater budget terms, and several in situ and satellite surface currents products were tested for advection, to identify processes driving SSS seasonal variations. With the same objective, Da-Allada et al. (2014) and (Camara et al., 2015) used slightly different tropical Atlantic configurations of the NEMO OGCM (Ocean general circulation model), where mixed-layer salinity budget terms were

computed online, on the model spatial grid (0.25°) and at each time step (20 min). We know from analyzing online diagnostics, that the nonlinear advective terms cannot be neglected. Observational data, however, often have lower temporal and/or spatial resolution than would be necessary to fully capture the nonlinear terms. Moreover, it is much more difficult to estimate subsurface vertical terms than surface terms with observations, which leads to a non-negligible residual term, which





interpretation is problematic. Online computation in an OGCM is practically the only way to close a mixed-layer heat/salt

budget.

In this paper, we use a joint observation-model approach: we compare the mixed-layer heat and salt budget terms estimated in a recently available observation-based climatological product to those simulated by a NEMO high-resolution simulation in the eastern tropical Atlantic. A sensitivity test to the spatio-temporal resolution at which advection terms are computed in the

model is conducted. This comparison should allow providing a high-level model validation, isolating the contribution of mesoscale advection in the mixed-layer budgets, and quantifying the uncertainty on the different budget terms. We particularly focus for the mixed-layer budgets on the upwelling regions where oceanic processes are expected to be dominant. The observational product, the model and the methodology used are presented in section 2. Section 3 contains the results of observation-model comparison regarding mean heat and salt budgets, and their seasonal variability in selected regions. In

section 4, discussions and conclusion are presented.

## 2 Data and methods

### 2.1 Data

#### 2.1.1 Observations

We use the PREFCLIM (PREFACE Climatology) observed seasonal climatology of mixed-layer heat and salt budgets

covering the eastern tropical Atlantic (Rath et al., 2016). It has been produced in the frame of the European PREFACE (Enhancing prediction of tropical Atlantic climate and its impacts) project, which aimed at improving climate models in the tropical Atlantic. This climatology is derived from all hydrographic data publicly available covering the region, including Argo floats data (Argo, 2000), and gliders measurement conducted by Geomar between 2002 and 2015 (for more details, see https://gliderweb.geomar.de), also completed by data from hydrographic stations in Senegal, Angola and Namibia waters

collected during cruises of the EAF-Nansen. In addition, this climatology uses data from other projects of PREFACE partners like PIRATA (Prediction and Research Moored Array in the Tropical Atlantic).

These data have been gridded using an interpolation scheme including isobaths-following and front-sharpening components (Schmidtko et al., 2013). Mixed-layer properties like temperature, salinity and depth are provided with a spatial resolution of 0.25° x 0.25°, while mixed-layer budget terms and horizontal velocities used to compute advection terms are given with a

lower spatial resolution of 2.5° x 2.5°. Surface heat fluxes are derived from the TropFlux data set (Kumar et al., 2012), the freshwater flux associated with evaporation is computed from latent heat flux of TropFlux and precipitation derived from GPCP (Global Precipitation Climatology Project) version 2.2 (Huffman et al., 2009). Heat/salt horizontal advection terms (split in zonal and meridional components) have been calculated using the near-surface gridded velocity field based on the measurements of surface drifters (Lumpkin et al., 2013) and ARGO floats surface drift from YOMAHA data set (Lebedev et



al., 2007), which is combined with a gridded temperature/salinity - gradient field. These will be called the gridded advection terms. This climatology was supplemented with additional product of heat/salt advection using estimated velocities from each of the drifter and float data points used for the gridded velocity fields combined with the full high-resolution hydrographic climatology. These alternative terms of heat/salt advection will be here called Lagrangian advection terms and denoted by Obs-drift, to highlight the difference with the previous terms. The full description of this climatology is presented in a PREFACE Project deliverable (Dengler and Rath, 2015).

### 2.1.2 Model

We use a regional configuration of the NEMO (Nucleus for European Modeling of the Ocean; Madec, G., 2014) oceanic model version 3.6. This regional simulation covers the tropical Atlantic (35°S - 35°N, 100°W - 15°E). It uses a horizontal Arakawa grid of type C, with a 0.25° horizontal resolution. The vertical grid, in z-coordinates, has 75 levels, including 12 levels within the upper 20 m and 24 in the upper 100 m of the ocean. The model is forced by daily outputs of the global MERCATOR reanalysis GLORYS2V3 at lateral boundaries. Atmospheric fluxes of heat, freshwater, and momentum used for surface forcing are from the Drakkar Forcing Set version 5.2 (DFS5.2) product (Dussin et al., 2016). The surface fluxes are prescribed following bulk formula (Large and Yeager, 2009). River runoffs are introduced as surface freshwater at river mouth and are based on a monthly climatology (Dai and Trenberth, 2002). Heat and salt budget terms are computed online, at each model time step (20 min), and vertically integrated in the mixed-layer. The mixed-layer depth is computed following a density criteria: a 0.03 kg m$^{-3}$ difference relative to the density at 10 m (de Boyer Montégut et al., 2004). The description of this model is more detailed in (Hernandez et al., 2016). In this paper, we use climatological monthly averages of mixed-layer properties averaged on the 1980-2015 period, to compare with similar terms available from observations. This climatological approach is used in many studies on mixed-layer budget (Da-Allada et al., 2014; Camara et al., 2015). This NEMO regional configuration has already been used to study the Gulf of Guinea salinity distribution and variability at seasonal and interannual timescales (Da-Allada et al., 2017; Awo et al., 2018).

### 2.2 Methods

In this paper, the driving processes of seasonal variability of mixed-layer temperature and salinity in selected regions are quantified through heat and salt budgets from NEMO model. This approach has been already used in several studies based on observations and models (Da-Allada et al., 2013; Hasson et al., 2013). In the following, as mixed-layer temperature and salinity are very close to SST and SSS, respectively, we indifferently use either vocabulary. The heat budget and salt budget evolution within the mixed-layer are respectively given by following equations (1) and (2), already used in previous studies (Peter et al., 2006; Jouanno et al., 2011; Da-Allada et al., 2014; Schlundt et al., 2014):

$$\frac{\partial SST}{\partial t} = \underbrace{\frac{Q^* + Q_s(1 - f_{z=-h})}{\rho_0 c_p h}}_{A} \underbrace{-\langle U.\partial_X T \rangle - \langle V.\partial_Y T \rangle}_{B} \underbrace{+ D_l(T)}_{C} \underbrace{-\langle W.\partial_Z T \rangle - \frac{(k_Z \partial_Z T)_{z=-h}}{h} - \frac{1}{h}\frac{\partial h}{\partial t}(SST - T_{z=-h})}_{D} \tag{1}$$





$$\frac{\partial SSS}{\partial t} = \underbrace{\frac{(E-P)SSS}{h}}_{A} \underbrace{-\langle U.\partial_X S\rangle - \langle V.\partial_Y S\rangle}_{B} \underbrace{+D_l(S)}_{C} \underbrace{-\langle W.\partial_Z S\rangle - \frac{(k_Z \partial_Z S)_{Z=-h}}{h} - \frac{1}{h}\frac{\partial h}{\partial t}(SSS - S_{z=-h})}_{D} + \underbrace{\frac{-R.SSS}{h}}_{A'}$$ (2)

Where $T$ is temperature and $S$ is salinity averaged within the mixed-layer of depth $h$, $(U, V, W)$ are zonal, meridional, vertical components of the velocity vector, and $D_l(.)$ is lateral diffusion. In equation (1), $Q^*$ and $Q_s$ represent the non-solar and solar components of surface heat flux respectively, and $f_{z=-h}$ represents the fraction of shortwave radiation reaching depths below the base of the mixed-layer (and hence not being available for heating up the mixed-layer itself).


The left-hand side of equation (1) represents the mixed-layer temperature tendency term, and the right-hand side represents all terms contributing to the heat budget. Namely, term $A$ is the surface heat flux decomposed (from left to right) in non-solar flux (long wave, latent heat, sensible heat) and solar flux (short wave). Term $B$ is horizontal temperature advection, decomposed in zonal and meridional components, term $C$ is lateral temperature diffusion, and term $D$ represents vertical oceanic processes.

$D$ contains (from left to right) vertical temperature advection, vertical temperature diffusion at the base of mixed-layer, and entrainment that represents mixed-layer temperature variations due to changes in the mixed-layer depth.

The left-hand side of equation (2) represents the mixed-layer salinity tendency term, the right-hand side represents all terms contributing to the salt budget. Namely, term $A$ is the ocean-atmosphere freshwater flux, which includes Evaporation ($E$) and

Precipitation ($P$). Term $B$ is horizontal salinity advection, decomposed in zonal and meridional components, term $C$ is lateral salinity diffusion, and term $D$ represents vertical oceanic processes. $D$ contains (from left to right) vertical salinity advection, vertical salinity diffusion at the base of mixed-layer and entrainment due to changes in the mixed-layer depth. The last term $A'$ represents the local river Runoff ($R$) contribution.

The budget computation slightly differs between the observation-based climatology and the model data. All terms are computed online in the NEMO model, explicitly from equations (1) and (2) except for the entrainment term that is estimated (using the online advection term) as a residual. In the PREFCLIM observed climatology, equations for the heat and salt budgets are simplified, as done in other studies (Stevenson and Niiler, 1983; Foltz et al., 2003, 2004; Delcroix and Henin, 1991; Schlundt et al., 2014). Only the tendency, surface heat or freshwater flux (term $A$) and advection terms (term $B$) are computed

explicitly, following equations (1) and (2). The residual is composed of unresolved vertical processes like diapycnal heat and salt fluxes, runoff contribution in the case of the salt budget, and accumulated errors from the explicitly resolved terms that can be due to sub-mesoscale (Dengler and Rath, 2015). For comparison between observations and model, terms $C$ and $D$ in equation (1), terms $C$, $D$ and $A'$ in equation (2) are grouped in a model pseudo-residual term equivalent to the observation residual. The daily NEMO model online budget terms, available on a 0.25° grid, are re-sampled at the lower spatial resolution

(2.5° x 2.5°) and time resolution (monthly) of observed budget terms for comparison. The online computation in the model means that the advection term includes high-frequency mesoscale activity. To remove this part and to mimic the resolution of





the gridded observations used for the PREFCLIM advection terms, horizontal heat and salt advection terms are also recomputed offline, following equations (1) and (2), with monthly model outputs of currents and temperature/salinity re-sampled at 2.5°- resolution. This approach has been used in another salinity budget in the tropical Pacific (Hasson et al., 2013).

In addition, a new pseudo-residual is inferred from equations (1) and (2) where advection is calculated offline and called offline pseudo-residual. In order to evaluate the consistency between the PREFCLIM and the NEMO mixed-layer budget climatology, we use common statistics: Root Mean Square Deviation (RMSD), standard deviation, spatial and temporal correlations (r), and summarize them using Taylor diagrams (Taylor, 2001).

## 3 Results

### 3.1 Mixed-layer properties

A preliminary validation of the NEMO regional simulation is done by comparing modeled and observed annual means of mixed-layer depth, of mixed-layer temperature and of mixed-layer salinity and standard deviation of the latter two. The model (Fig. 1b) reproduces the large-scale properties of observed MLD (Fig. 1a) in the eastern tropical Atlantic. In both, observations and model, the shallowest mixed-layer is found along the Equator and the coasts of Africa, while the deepest mixed-layer is

found towards the northern and southern subtropical gyres. The main differences between the modeled and observed MLD are found along the northern coast of the Gulf of Guinea and along 24°S where the model MLD is shallower and along 12°S where the model MLD is deeper (see Figure A.1). Also, the MLD spatial variations are smoother in the observed product than in the model, despite a similar 0.25° spatial resolution. This is likely due to the fact that the observations under-estimate spatial variability because they are not available at the necessary resolution.

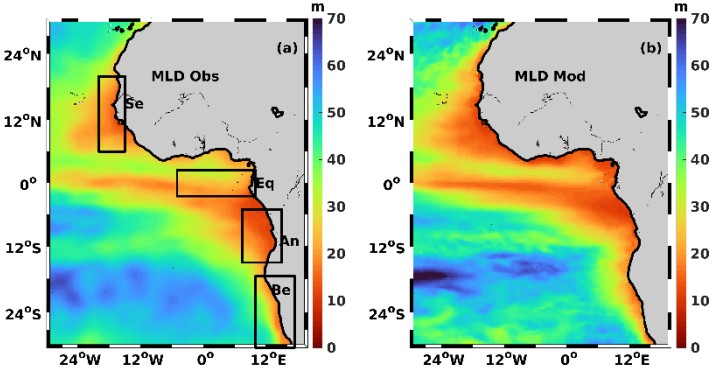


**Figure 1:** Annual mean of mixed-layer depth from observations (a) and model (b). The rectangles in (a) correspond to the 4 regions of study.

The model (Fig. 2b) reproduces well the observed mean SST (Fig. 2a) in the eastern tropical Atlantic. Highest SST is found along the zonal band between 0° and 12°N, while lowest SST is found in the Benguela and Canary Current regions south of

12°S and north of 18°N respectively, which are characterized by Eastern Boundary Upwelling Systems (EBUS, Chavez and




Messié, 2009). The coastal cooling in the Benguela region is weaker in the model. The cooling associated with the smaller coastal upwelling region north of the Gulf of Guinea (Djakouré et al., 2014, 2017) appears in the model only. Model and observations show similar large-scale patterns of SST seasonal variability (Fig. 2c-d) with an open ocean minimum between 12°S and 12°N, at the coast around 24°S and 24°N, and maximum variability in the open ocean south of 24°S, at the coast

around 15°N and 12°S. However, the model shows a smaller coastal seasonal variability of SST at 12°S compared to the PREFCLIM climatology, a larger variability in the open ocean south of 24°S, and captures the variability associated with the coastal upwelling north of the Gulf of Guinea that does not appear in PREFCLIM. These differences are probably associated with the strengths and weaknesses in the two data sources. On one hand, there is variable observational coverage, from the very densely sampled EAF-Nansen data along the shore to generally lower observational coverage in the regions away from

the shores. On the other hand, the wind forcing used for the model may not allow to fully capture the near-shore variability (Junker et al., 2015).

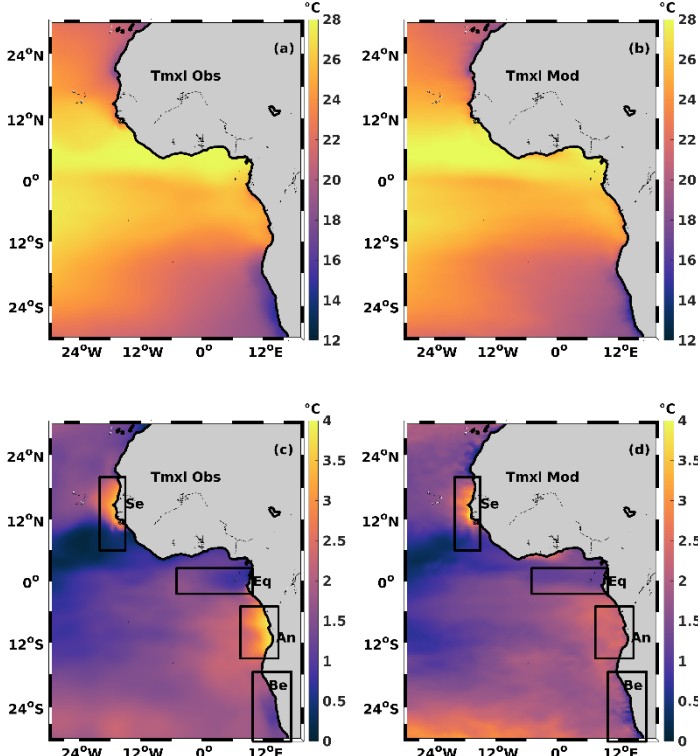

**Figure 2:** Annual (a and b) and standard deviation (c and d) of mixed-layer temperature from observations (left: a and c) and model (right: b and d). The rectangles in the bottom panels correspond to the 4 regions of study.


The model (Fig. 3b) also represents the main observed features of SSS (Fig. 3a) in the eastern tropical Atlantic. Highest SSS is found towards the center of, both, the north and the south subtropical gyres. On the opposite, the lowest SSS is found slightly north of the equator and in the Gulf of Guinea due to the strong precipitation associated with the ITCZ and river runoff. SSS



is also relatively low in the Benguela upwelling region. SSS is lower in the model than in observations in the eastern part of
the Gulf of Guinea, but higher along 12°S (see Figure A.1). Except for the Benguela region, the model (Fig. 3d) shows a strong
SSS variability associated with low SSS regions. However, the PREFCLIM climatology (Fig. 3b) shows a much weaker
variability than the model in the Gulf of Guinea, and around the Niger River in particular. This may be due to a poor temporal
resolution of SSS observations available here. Off the Niger River mouth, the model is in better agreement with other
observation-based SSS products showing a strong (~2 psu) variability here (Da-Allada et al., 2014).

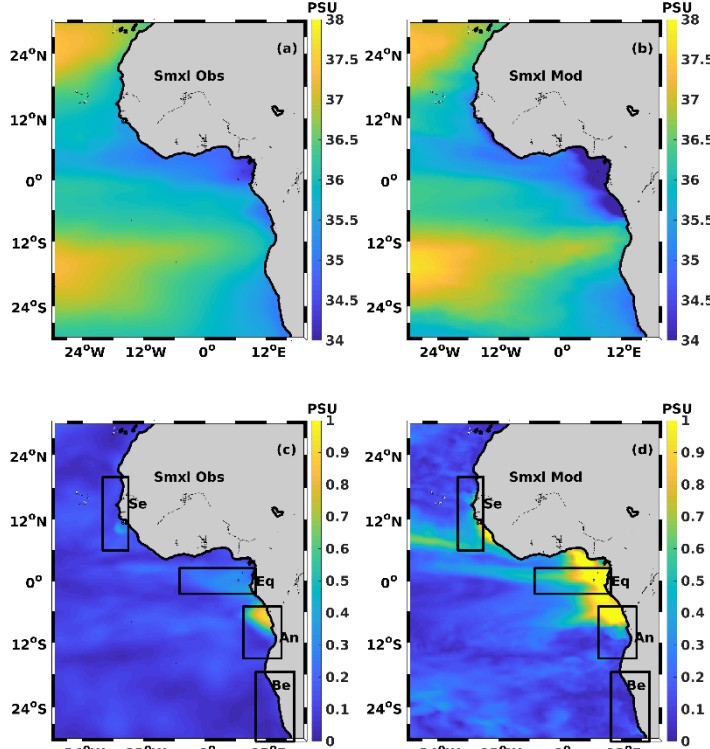


**Figure 3:** Annual mean (a and b) and standard deviation (c and d) of mixed-layer salinity from observations (left: a and c) and model (right:
b and d). The rectangles in the bottom panels correspond to the 4 boxes of study.

In the following part, the heat/salt budgets will be investigated in detail for four boxes selected for their particularly low mean
SST/SSS and/or strong SST/SSS variability. These are, from north to south, the Senegal box (6°N - 20°N, 20°W - 15°W), the
equatorial box (2.5°S - 2.5°N, 5°W - 10°E), the Angola box (15°S - 5°S, 7.5°E - 15°E) and the Benguela box (30°S – 17.5°S,
10°E – 17.5°E) (Fig. 1-3).





### 3.2 Mixed-layer heat budget

#### 3.2.1 Time-averaged spatial variations

We compared the mean mixed-layer heat budget from model and observations. The surface heat flux and horizontal heat advection maps are presented in Figure 4. Surface heat flux from observations is positive everywhere in the eastern tropical Atlantic, with a maximum along the equator that gets the strongest solar flux, and along the west coasts of Africa (Fig. 4a). Along west-African coasts, the heat flux is strong as solar flux can concentrate in a thin mixed-layer (Fig. 1), notably due to the strong salinity stratification induced by Niger and Congo rivers in the Gulf of Guinea. In addition, the temperature

difference between the ocean cooled by coastal upwelling (Benguela, Senegal, see Fig. 2) and the atmosphere leads to a reduced latent heat flux. The model reproduces the observed patterns with higher resolution (Fig. 4c) and, when re-sampled similarly (Fig. 4b), shows a good spatial agreement with the PREFCLIM climatology (r = 0.85). The seasonal variations of the NEMO and PREFCLIM heat fluxes are also very well correlated except along the equator (Fig. 4d). However, the NEMO model flux is biased low compared to PREFCLIM. It shows a net flux towards the atmosphere along zonal bands around 6°N and 12°S.

These differences can be explained by the different data sources used for surface heat flux in the PREFCLIM climatology (TropFlux) and as forcing for the NEMO model (DFS5.2), and may be also due to differences in the bulk formulae used to estimate fluxes in the model and in TropFlux.

As expected, there are important differences between the maps of offline heat advection calculated based on the coarsened model currents and hydrography (Fig. 4f) and the online heat advection taking into account the full spatio-temporal variability

(Fig. 4g, re-sampled to 2.5° after calculating advection). In the online version, advection acts to cool the mixed-layer in a thinner west-equatorial band and to mainly warm the mixed-layer in a larger part of the Gulf of Guinea. The offline advection is in much better agreement with the PREFCLIM climatology than the online advection (spatial correlation coefficient r = 0.54 vs r = 0.08). In PREFCLIM (Fig. 4e), the distributions of horizontal heat advection and surface heat flux are approximately opposite each other, with advection acting to cool the mixed-layer along most of the 3°S-3°N band and along the coast. This

sign is expected off the EBUS of Senegal and Benguela as temperature decreases towards the coast (eastward) and the zonal circulation is dominated by the westward Northern and Southern Equatorial Currents (NEC/SEC). The seasonal variations of the PREFCLIM and the NEMO offline advection terms show correlations quite different from one place to another and moderate on average but rather large slightly south of the equator (Fig. 4h). We also compared the annual mean spatial distribution from the two versions of model advection to the Lagrangian advection, and found that the offline advection is also

much better correlated than the online advection (r = 0.58 versus r = 0.37) with the Lagrangian advection terms (see Figure A.2 for more details).





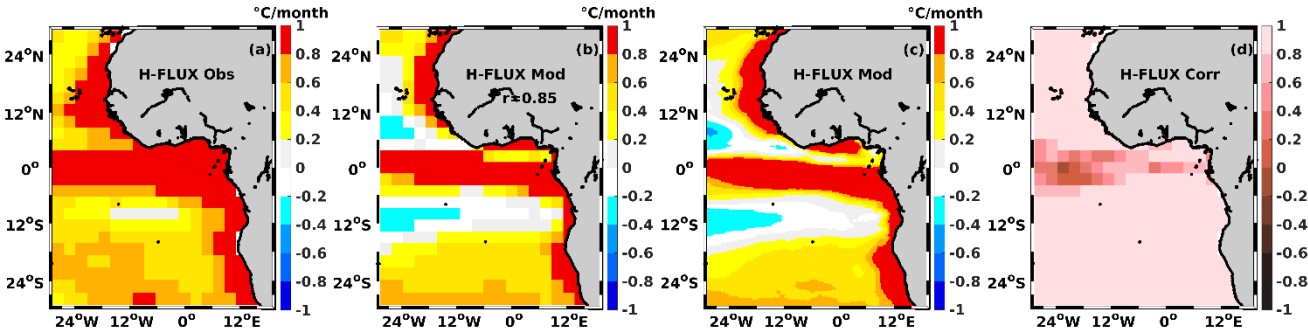

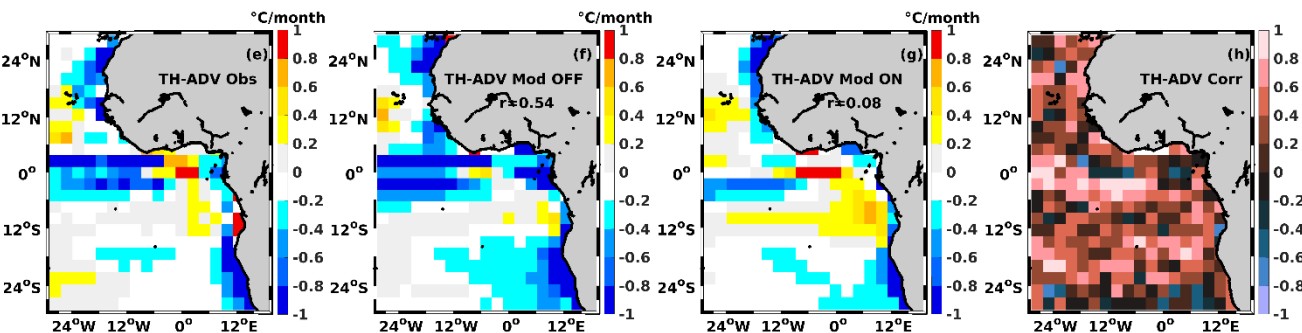

**Figure 4:** Mean heat flux from PREFCLIM (a), NEMO re-sampled at PREFCLIM 2.5° resolution (b) or at original 0.25° resolution (c), and seasonal correlation between PREFCLIM and NEMO heat flux (d). Mean horizontal heat advection from PREFCLIM (e), NEMO offline (f) or online computation re-sampled to 2.5° resolution (g), and seasonal correlation between PREFCLIM and NEMO offline advection (h). r in (b), (f) and (g) indicates the spatial correlation between PREFCLIM and NEMO, which is 95% significant when r > 0.12. The temporal correlation of seasonal cycles in (d) and (h) is 95% significant when r > 0.58.

### 3.2.2 Regional seasonal budget

In this part, we analyze the individual contributions of different physical processes to the heat budget during a seasonal cycle in selected regions. We present the seasonal variability of mixed-layer temperature and of the heat tendency term (Fig. 5), and try to identify the dominant processes. Taylor diagrams are used to evaluate the consistency of the global terms of the budget between PREFCLIM climatology and NEMO model (Fig. 6). In the following, the observed gridded advection, rather than the observed Lagrangian advection, is used because it is generally better correlated with the model advection (see Figures A.3 and A.4 for more details).

In the Senegal region, observed and modeled seasonal mixed-layer temperature variations (Fig. 5a) are largely consistent (r = 0.96, RMSD = 0.64 °C). They both show an annual cycle with a SST maximum around 27 °C in September and a minimum of around 22°C in the middle of the October-May upwelling season (Ndoye et al., 2014). The minimum is however found one





month later in the model (March) than in the observations (February). The related tendency terms from observations and model
(Fig. 5a) agree to a lesser extent (r = 0.91 and RMSD = 0.43 °C per month) which is mainly present in May-June when the
modeled warming is larger than the observed warming. The PREFCLIM and the modeled regional heat budgets (Fig. 5b) agree
on the seasonal variations of heat flux (r = 0.97 and RMSD = 0.24 °C per month). Differences between observations and model
are larger for horizontal advection, and again offline advection from NEMO compares slightly better than online advection
with PREFCLIM (r = 0.52, RMSD = 1.34 °C per month vs r = 0.46, RMSD = 0.99 °C per month, see Figure A.3 for more
details). The (pseudo-) residual terms however compare better for online advection rather than for offline advection (r = 0.96
and RMSD = 0.72 °C per month vs r = 0.51 and RMSD = 0.87 °C per month, respectively). Overall, in the Senegal region, the
seasonal cycle of mixed-layer temperature is mainly controlled by surface heat fluxes that are strong and drive the warming
from March to September. For the rest of the year, they are small or negative and, with the help of horizontal advection and
vertical diffusion, induce cooling (Fig. 5c).


In the Equatorial region, Fig. 5d presents the seasonal evolution of mixed-layer temperature in model and observations. There
is a strong consistency (with r = 0.98 and RMSD = 0.28 °C) between the two terms, with a SST minimum in the same month
of August while the model reaches the SST maximum in April, one month after the observations. Temperature tendency terms
are also consistent (r = 0.94 and RMSD = 0.27 °C per month) although the maximum cooling in May is strongest in the model.
We note also a relatively good consistency in the seasonal cycle of the heat flux (Fig. 5e, r = 0.78 and RMSD = 1.14 °C per
month). There is much less agreement between NEMO and PREFCLIM for the horizontal advection term, whether it is
computed online (r = 0.15 and RMSD = 1.37 °C per month) or offline (r = -0.69 and RMSD = 4.27 °C per month) in the model.
The observed residual term compares much better with the model online pseudo-residual term (r = 0.83 and RMSD = 0.56 °C
per month) than with the offline pseudo-residual (r = 0.11 and RMSD = 1.15 °C per month). Overall, in this Equatorial box,
according to the model the seasonal cycle of mixed-layer temperature is essentially driven by vertical heat diffusion (Fig. 5f)
as its variations are rather similar as those of the temperature tendency term, except for a shift that can be explained by the
heat flux, which remains positive and relatively constant all year long.

In the Angola region, the model reproduces well the seasonal evolution of observed mixed-layer temperature (Fig. 5g, r = 0.96
and RMSD = 0.86 °C). The maximum SST observed in March is lagged by one month in the model but the minimum SST in
found in August in both NEMO and the PREFCLIM climatology. Differences are relatively larger for the heat tendency term
(r= 0.84 and RMSD = 0.80 °C per month) that is in the model smaller than observed in November-December (Fig. 5g). We
note a strong agreement between modeled and observed heat flux (Fig. 5h, r = 0.97 and RMSD = 0.24 °C per month). On the
contrary, horizontal heat advection terms are poorly correlated when computed online in the model (r = 0.05 and RMSD =
1.04 °C per month), but we observe an improvement when computed offline (r = 0.44 and RMSD = 0.90 °C per month). This
suggests an important role of nonlinear terms in the heat budget. The resulting observed residual term and model (online and
offline) pseudo-residuals term are also quite different (respectively r = 0.51 and RMSD = 0.90 °C per month and r = 0.44 and



RMSD = 0.89 °C per month). In the Angola box, the seasonal cycle of the mixed-layer heat budget is mostly controlled by heat fluxes, especially solar fluxes, with also contribution from zonal advection and vertical diffusion (Fig. 5i).


In the Benguela region (Fig. 5j), the model reproduces well the observed seasonal cycle of the mixed-layer temperature, which is maximum in February-May and minimum from July to October, but with a positive bias close to 1 °C (r = 0.97, RMSD = 0.82 °C). The heat tendency term of the model agrees with observations (Fig. 5j, r = 0.95 and RMSD = 0.25 °C per month). Heat flux variations are very well correlated (Fig. 5k, r = 0.99, RMSD = 1.16 °C per month). Horizontal heat advection

variations are moderately correlated for offline computation (r = 0.70, RMSD = 0.90 °C per month), even for online computation (r = 0.68, RMSD = 1.11 °C per month), and the observed residual and modeled pseudo-residual are also consistent (r = 0.73, RMSD = 0.68 °C per month vs r = 0.60, RMSD = 0.82 °C per month for online/offline version respectively). In the Benguela region, the heat budget seasonal cycle is largely controlled by heat flux warming, balanced by the cumulative cooling effect of zonal heat advection and vertical diffusion (Fig. 5l).

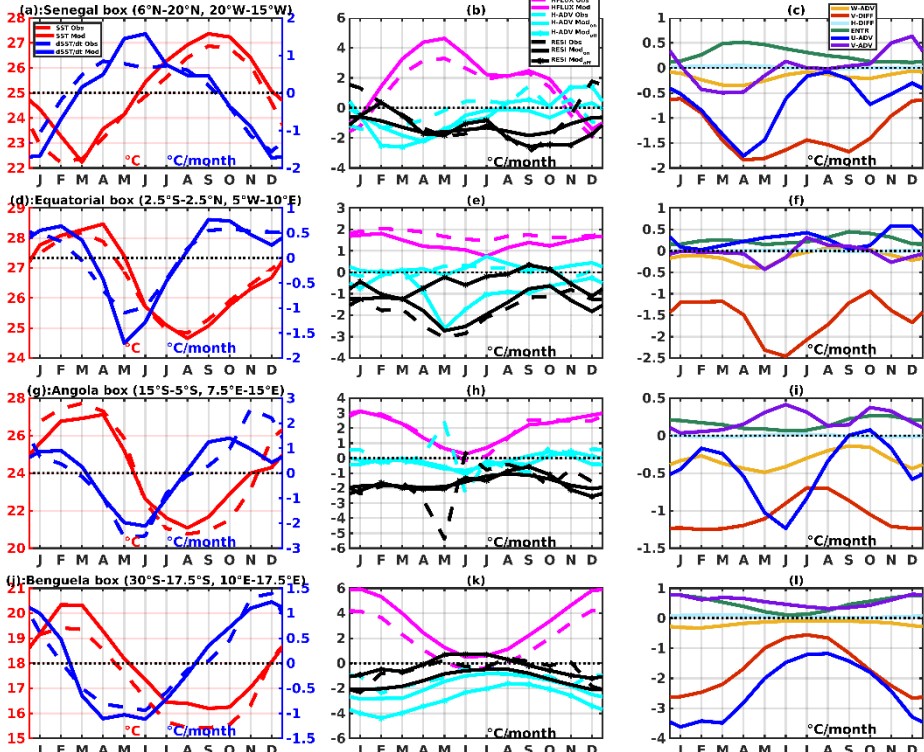


**Figure 5:** Seasonal mixed-layer heat budget terms from observations (dashed line) and model (full line and full dotted line for pseudo-residual associated with offline advection) in selected regions: SST in °C and tendency terms in °C per month.





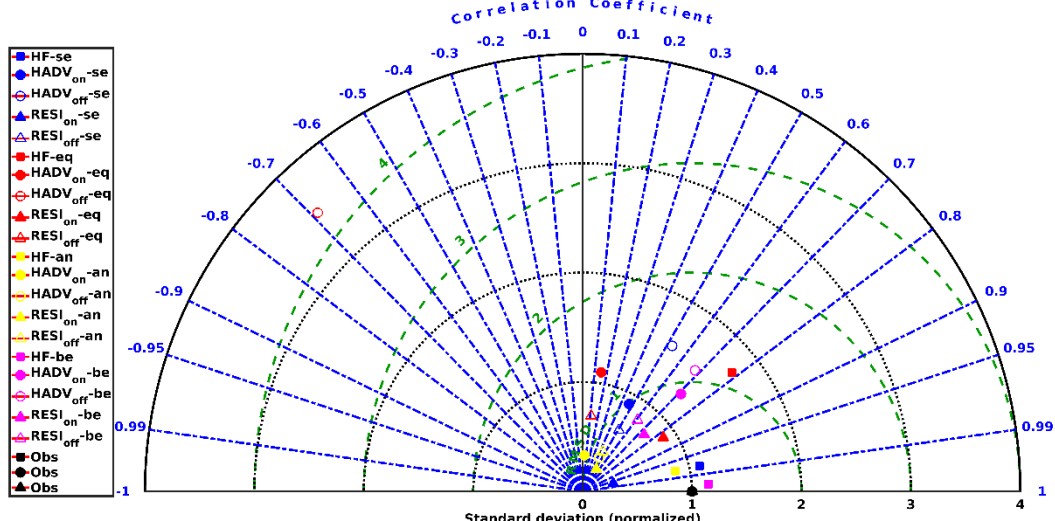

**Figure 6:** Taylor diagram of global terms of heat budget in selected regions. Heat flux, horizontal advection (gridded advection for observations and online advection for model), and (pseudo-) residuals are represented by squares, circles, triangles respectively. Empty circle and triangle are offline advection and associated (pseudo-) residuals. Senegal, Benguela, Equatorial and Angola regions are designed by blue, red, yellow and magenta colors respectively. Correlations are 95% significant when r > 0.58.

## 3.3 Mixed-layer salt budget

### 3.3.1 Time-averaged spatial variations

We now compare the mean salt budget from model and from observations through freshwater flux and horizontal salt advection (Fig. 7). The PREFCLIM freshwater flux acts to decrease the mixed-layer salinity along the 0-12°N equatorial band, dominated by strong precipitation due to the ITCZ (Fig. 7a). Elsewhere in the eastern tropical Atlantic, freshwater flux is dominated by evaporation, which acts to increase mixed-layer salinity, a feature of the north and south subtropical gyres. Evaporation is maximum off Angola, where southern trade winds can enhance it. The model freshwater flux forcing reproduces these patterns and, when re-sampled to the PREFCLIM resolution (Fig. 7b), shows a good spatial agreement with the PREFCLIM climatology (r = 0.96). However, the NEMO freshwater flux shows a negative bias along the 6°N - 12°N band and in the Gulf of Guinea, and a high positive bias in the rest of eastern tropical Atlantic basin. These differences are likely due to the different data sources used for surface freshwater flux in the PREFCLIM climatology (TropFlux for evaporation and GPCP for precipitation) and as forcing for the NEMO model (DFS5.2). However, the seasonal variations of freshwater flux are generally well correlated between NEMO and PREFCLIM except for a few regions (Fig. 7d). As expected, there are important differences between the maps of offline salt advection (Fig. 7f) and online salt advection (Fig. 7g, re-sampled at 2.5°). In the online version, advection strongly acts to decrease mixed-layer salinity almost everywhere in the eastern tropical Atlantic. However, we note that in the offline version (Fig. 7f) and in observations (Fig. 7e), advection acts to increase salinity for some regions: in the 4°N - 10°N equatorial band, west of 12°W between 30°S - 18°S, off Angola and in the northern Gulf of Guinea.



The spatial distribution of offline advection is in much better agreement with the PREFCLIM climatology than online advection (r = 0.48 vs r = 0.17). Schematically, in the eastern tropical Atlantic, salinity increases poleward from the equator, due to strong evaporation in the subtropical gyres, which drives the meridional gradient (Fig. 3a-b). Besides, salinity decreases toward the east, because of freshwater intakes from rivers in the Gulf of Guinea, resulting in a westward increase of SSS. In the Gulf of Guinea, the observed saltening by advection can be explained by this SSS gradient transported by the eastward

southern Guinea Current (GC), following Equation 1. The freshening by advection in a large part of the eastern tropical Atlantic basin is expected as the circulation is, on the contrary, dominated by the westward NEC/SEC. In addition, the presence of alongshore currents like the southward Angola current and northward Benguela current in these coastal regions can drive either a salting or a freshening due to horizontal salt advection, as shown with NEMO. Moreover, there can be competition between zonal and meridional salt advection (Da-Allada et al., 2013). The correlation between seasonal cycles of advection in NEMO

and PREFCLIM is very dependent on the location but it is generally stronger in the Gulf of Guinea and toward the subtropical gyres (Fig. 7h). We also compared the annual mean spatial distribution from the two versions of model advection to the Lagrangian advection and found that the offline advection is also much better correlated than the online advection (r = 0.51 versus r = 0.15) with the Lagrangian advection. This suggest the importance of non-linear terms. More statistics on comparison of zonal/meridional advection terms can be found in Figure A.3.

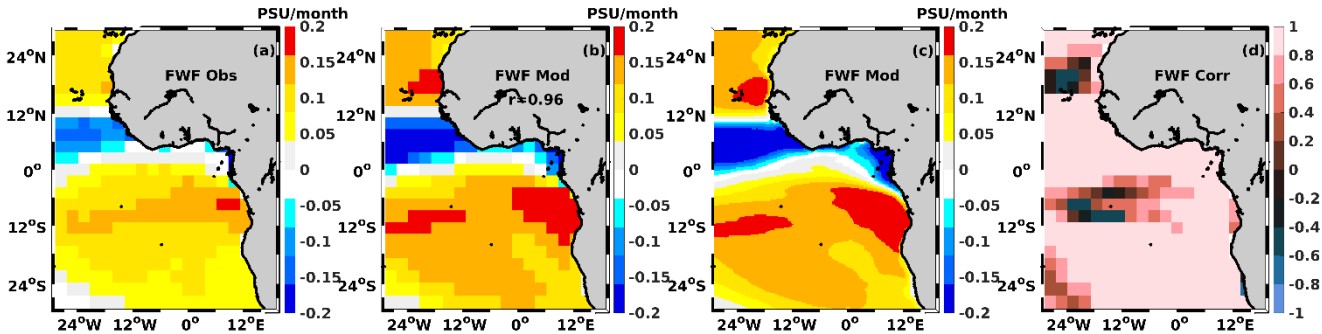

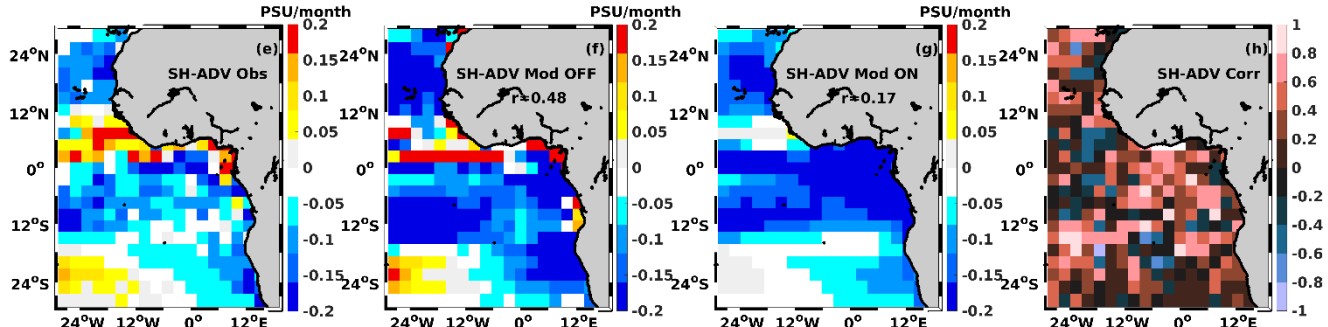


**Figure 7:** Mean freshwater flux from PREFCLIM (a), NEMO re-sampled at PREFCLIM 2.5° resolution (b) or at original 0.25° resolution (c), and seasonal correlation between PREFCLIM and NEMO freshwater flux (d). Mean horizontal salt advection from PREFCLIM (e),





NEMO offline (f) or online computation re-sampled at 2.5° resolution (g), and seasonal correlation between PREFCLIM and NEMO offline advection (h). R in (b), (f) and (g) indicates the spatial correlation between PREFCLIM and NEMO, which is 95% significant when r > 0.12.

The temporal correlation of seasonal cycles in (d) and (h) is 95% significant when r > 0.58.

### 3.3.2 Regional seasonal budget

As previously done for the heat budget (see Figures A.5 and A.6 for more details), we evaluate the individual contributions of different physical processes to the salt budget during a seasonal cycle (Fig. 8), and try to identify the dominant processes. Taylor diagrams are used to evaluate the consistency of budget terms between PREFCLIM climatology and NEMO model

(Fig. 9).

In the Senegal region, observed and modeled mixed-layer salinity seasonal cycles (Fig. 8a) are very different (r = 0.59, RMSD = 0.33 psu). SSS variations are around 0.1 psu during the seasonal cycle in observations, but reach 0.7 psu in the model, with a maximum in May and a minimum in October (the latter also seen in observations). The SSS increase from October to May

coincides with the upwelling season (Ndoye et al., 2014). The seasonal variations of the tendency salinity terms (Fig. 8a) are also quite different (r = 0.55, RMSD = 0.12 psu per month). The observed term remains weaker than the modeled term, they vary in opposite phase from December to March, then in phase from this month on, but the modeled freshening is larger than observed in May-November. However, the seasonal variations of freshwater flux (Fig. 8b) show good agreement between observations and model (r = 0.99, RMSD = 0.39 psu per month). Modeled and observed horizontal salt advection are less

correlated (r = 0.51, RMSD = 1.34 psu per month for online vs r = 0.57, RMSD = 3.87 psu per month for offline), and there are very large differences (r = -0.19, RMSD = 1.65 psu per month for online vs r = -0.15, RMSD = 4.09 psu per month for offline) between the observed residual term and the model pseudo-residual term. In this region, the balance is controlled in large part by freshwater flux because of the compensation between different oceanic processes. From June to October the observed freshening can be explained by precipitation, which reaches its maximum between July and August because of the

ITCZ position over Sahel, and associated runoff particularly from the Senegal and Gambia rivers. From October to November, the freshening is associated with the combined effect of zonal advection (Fig. 8c), precipitation and river runoff, in this order. For the rest of the year, evaporation plays a dominant role and increases mixed-layer salinity.

In the Equatorial region, the modeled and the observed seasonal cycle of the mixed-layer salinity are largely in phase

(Fig. 8d, r = 0.82, RMSD = 0.79 psu) but the modeled salinity has a seasonal cycle 3 times stronger and is lower by almost 1.5 psu between April and May. This could be due to the fact that the observations miss some strong and very shallow near-surface stratification, which is averaged into the surface grid box of the model. The minimum SSS observed in February is lagged by two months in the model but the maximum SSS is found in October in both NEMO and the PREFCLIM climatology. The related tendency terms (Fig. 8d) are weakly correlated (r = 0.58, RMSD = 0.27 psu per month). The model term has a stronger

amplitude throughout the cycle with some peaks in January, May and November. Fig. 8e shows that freshwater flux are



strongly correlated (r = 0.94, RMSD = 0.90 psu per month). The horizontal advection terms are quite different when online computation is used (r = 0.38, RMSD = 4.03 psu per month) but compare better with the offline advection (r = 0.77, RMSD = 8.56 psu per month), although the modeled advection show in both cases stronger variations than observed. The model pseudo-residual also compares better with the observed residual for the offline version than for the online version (r = 0.81, RMSD =

3.51 psu per month vs r = 0.54, RMSD = 1.76 psu per month respectively). In the Equatorial region, the seasonal variability of mixed-layer salinity is mostly due to vertical salt diffusion and zonal salt advection (Fig. 8f). From September to March, zonal salt advection increases mixed-layer salinity. Vertical salt diffusion plays a major role to increase mixed-layer salinity the rest of year, in particular in May when it reaches his maximum. The contribution of freshwater flux, vertical and meridional salt advection are weak and can compensate each other during the seasonal cycle.


      In the Angola region, Fig. 8g presents the seasonal evolution of mixed-layer salinity in model and observations. The model reproduces relatively well (r = 0.55, RMSD = 0.28 psu) the seasonal evolution of observed SSS from the beginning of the year to August. In June, both the observed and modeled SSS are around their maximum, but SSS in the model decreases then progressively until it reaches its minimum in November, when the observed SSS only begins to decrease until its minimum

in February, the month where the model reaches his second minimum. The related tendency terms present a disagreement above all at the end of cycle (Fig. 8g, r = 0.29, RMSD = 0.27 psu per month). SSS in NEMO reaches its maximum in May and its minimum in February, while observed SSS reaches its maximum in April and its minimum in December when the model shows a secondary maximum. Surface freshwater fluxes are both positive all year long in NEMO and PREFCLIM, indicating evaporation is stronger than precipitation, with similar variations (Fig. 8h r = 0.79, RMSD = 0.66 psu per month). In addition

to these freshwater fluxes, there is strong runoff associated with the freshwater discharge from the Congo River that flows in this box, which is a major driver of SSS here (Houndegnonto et al., 2021), included in NEMO only. Horizontal advection in model and observations are quite different (r = 0.15, RMSD = 2.13 psu per month for online version vs r = 0.09, RMSD = 2.51 psu per month for offline version), and the modeled pseudo-residual compares better with the observed residual for the online version (r = 0.64, RMSD = 0.78 psu per month) than for the offline version (r = 0.38, RMSD = 1.43 psu per month). According

to the model, in this region, the salt budget seasonal cycle is mostly driven by oceanic processes, namely meridional advection, vertical diffusion and advection, in this order (Fig. 8i).

      In the Benguela region, the model follows well the observed seasonal cycle of mixed-layer salinity (r = 0.76, RMSD = 0.06 psu), though with a negative mean bias (around -0.06 psu) throughout the cycle (Fig. 8j). The modeled SSS reaches its

maximum in May, one month later than observed, and its minimum in December, also one month later than observed. The salt tendency term of the model also reproduces relatively well the observed term (r = 0.68, RMSD = 0.01 psu per month), especially from January to July (Fig. 8j). The model shows a maximum SSS increase in February, one month earlier than observed, and maximum SSS decreases in August and December, 2 months later than the observed peaks. NEMO and PREFCLIM freshwater fluxes are consistent (Fig. 8k, r = 0.92, RMSD = 0.88 psu per month). Horizontal advection terms are





less consistent between observations and model (r = 0.56, RMSD = 0.83 psu per month for online version vs r = 0.15, RMSD = 1.04 psu per month for offline version), as pseudo-residual terms (r = 0.50, RMSD = 0.92 psu per month for online version vs r = -0.09, RMSD = 1.10 psu per month for offline version). In the Benguela region, the salt budget seems to be mostly controlled by freshwater flux and zonal salt advection. The observed freshening from March to December can be partly explained by the minimum of evaporation in June, followed by the increasingly negative effect of zonal advection.

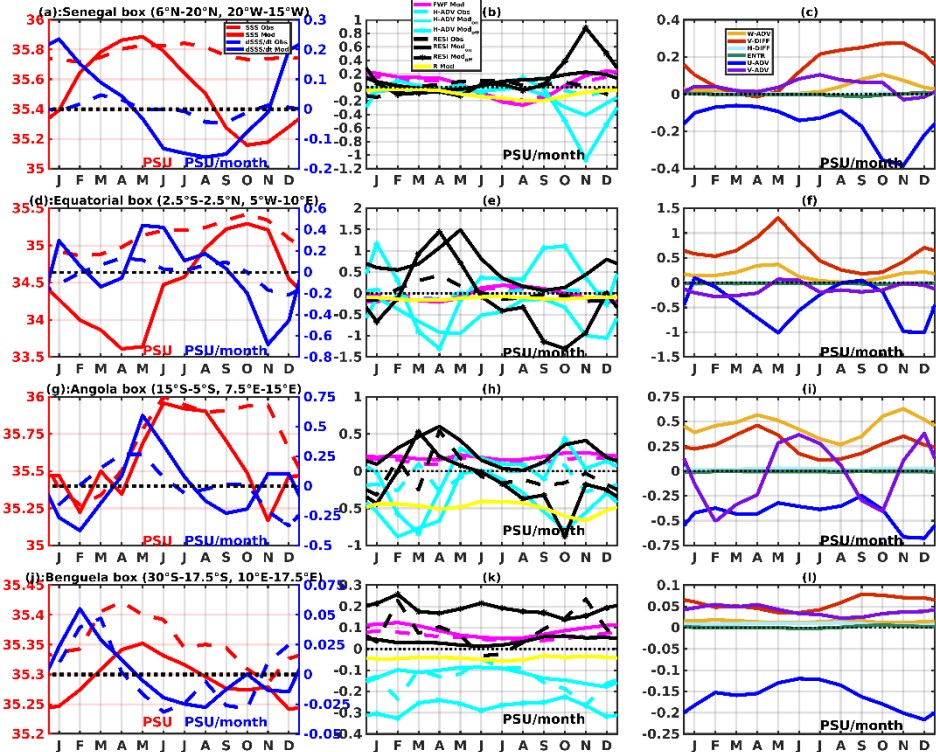


**Figure 8:** Seasonal mixed-layer salt budget terms from observations (dashed line) and model (full line and full dotted line for pseudo-residual associated with offline advection) in selected regions: SSS in psu and tendency terms in psu per month.





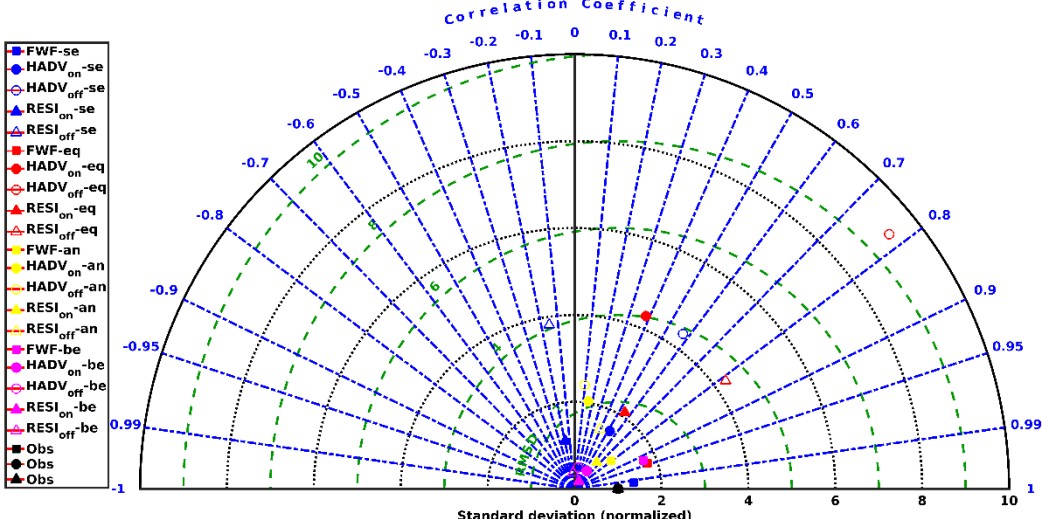

**Figure 9:** Taylor diagram of global terms of salt budget in selected regions. Freshwater flux, horizontal advection (gridded advection for observations and online advection for model), and (pseudo-) residuals are represented by squares, circles, triangles respectively. Empty circles and triangles are offline advection and associated (pseudo-) residuals. Senegal, Benguela, Equatorial and Angola regions are designed by blue, red, yellow and magenta colors respectively. Correlations are 95% significant when r > 0.58.

## 4 Discussion and conclusions

In this paper, we examined the dominant physical processes controlling the seasonal variability of mixed-layer heat and salt budgets in selected coastal regions of the eastern tropical Atlantic, namely the Senegal, the Equatorial, the Angola, and the Benguela regions. First, we used both a regional configuration of the NEMO model and the PREFCLIM observation-based climatology to analyze the spatial variations of the annual mean mixed-layer heat and salt budgets in the eastern tropical Atlantic (see Fig. 4 and Fig. 7 respectively). The model outputs were re-sampled to the PREFCLIM time/space resolution to compare maps of the mean processes contributing to mixed-layer heat/salt budgets, according to both sources. Second, we analyzed the seasonal variation of the mixed-layer temperature and salinity, their related tendencies and potential driving processes: heat/freshwater flux, horizontal heat/salt advection, and others processes estimated from observations as a residual but explicitly resolved in the model, in the selected regions. As the PREFCLIM climatology does not capture the mesoscale physical processes, we relied on the high-resolution model outputs to evaluate their contribution to the mixed-layer heat/salt budget.

For the preliminary validation, the results have shown that the model reproduces consistently the mean features of observed mixed-layer depth, temperature, and salinity in the eastern tropical Atlantic (see Fig. 1, Fig. 2 and Fig. 3 respectively). The existing differences between modeled outputs and the PREFCLIM climatology can be explained by the different heat and salt



flux products that are used for forcing the model or for estimating the PREFCLIM budget terms. There are also differences in
the method to define the MLD. The PREFCLIM climatology uses the algorithm of Holte and Talley (2009) whereas the model
uses the density criterion (0.03 kg m$^{-3}$ relative to the density at 10 m depth) recommended by de Boyer Montégut et al. (2004).
For example, the observed strong positive bias of mixed-layer salinity in the model relative to observations around 12°S is
associated to a positive bias in MLD. Except for these differences, the model and the PREFCLIM climatology capture the
shallow near-shore mixed-layer depth along the equator and the western coast of Africa where our selected regions are
localized.

For the secondary validation, we have used both the model and the PREFCLIM climatology to analyze the annual mean of
heat/freshwater flux and horizontal heat/salt advection. The model heat/freshwater fluxes largely agree with the PRECLIM
climatology (Fig. 4a-b and Fig. 7a-b respectively), except for differences in a few regions that can again be due to different
flux products or MLD biases. There are important differences between the model heat/salt advection terms computed either
offline (Fig. 4f and Fig. 7f respectively), at PREFCLIM spatio-temporal resolution (2.5°, monthly) or online (Fig. 4g and Fig.
7g respectively), at high spatio-temporal resolution (0.25°, 20 min) with subsequent monthly re-sampling at 2.5°. These
differences are explained by the high-frequency variability related to mesoscale and sub-mesoscale dynamics, which is
included in the second case but not in the first case.

At seasonal time scales, the monthly mixed-layer heat/salt tendency terms in the selected regions are very weak in, both, the
PREFCLIM climatology and the model, compared to individual terms contributing to the heat/salt budgets that tends to
compensate each other, as also found in previous studies (Da-Allada et al., 2013, 2014; Camara et al., 2015).

Surface heat fluxes, and especially the solar flux, dominate the seasonal mixed-layer heat budget in the Senegal, the Angola
and the Benguela regions (Fig. 5b, h, and k respectively). In the Senegal region, this result, and the secondary contribution of
oceanic processes such as vertical diffusion and zonal advection, which add to latent heat flux to drive the observed winter
cooling, confirm previous studies (Carton and Zhou, 1997; Yu et al., 2006).

In the Equatorial region, the heat flux remains positive and near constant throughout the seasonal cycle. This shows the
dominance of the shortwave flux that warms the mixed-layer from September to April, although this warming weakens
between November and December. Although our selected box slightly differs with previous regional studies, this result is in
agreement with earlier studies (Peter et al., 2006; Wade et al., 2011). The variability of mixed-layer temperature, in particular
the observed spring/summer cooling during the formation of the ACT, is mainly controlled by vertical heat diffusion (Fig. 5e),
confirming other studies (Yu et al., 2006; Jouanno et al., 2011). While it does not compensate for the cooling effect of vertical
diffusion, zonal heat advection is positive all year long in the equatorial region, the only one among analyzed regions where it
is so. This is the consequence of a negative zonal temperature gradient as the mixed-layer temperature decreases toward the
coast, advected by westward currents associated with the SEC. When associated with meridional heat advection, this leads to
a positive horizontal heat advection throughout all year but in the month of May. We note a near similar variability of horizontal
advection in Wade et al. (2011), although this term is negative in their study except for June-July, when we both observe a





positive maximum. This difference can be linked either to products used or to the criterion used to define the MLD (temperature vs density criterion), and maybe to the slightly different boxes of study. Jouanno et al. (2017) also found, like us, a permanent

warming effect of horizontal advection in an equatorial box shifted west compared to ours, using the same model configuration. There is overall agreement on the dominant role of vertical mixing to cool the mixed-layer during ACT formation. This vertical mixing due to vertical diffusion is explained by the strong vertical shear between the Equatorial Under-Current (EUC) and the SEC in our selected region, as discussed in previous studies (Wade et al., 2011; Jouanno et al., 2011; Hummels et al., 2013; Schlundt et al., 2014).

In the Angola region, the dominant role of heat flux in the mixed-layer heat budget was also found in previous studied (Carton and Zhou, 1997; Yu et al., 2006). In this region, the incoming shortwave flux warms the mixed-layer from August until Mars against the action of latent heat flux. The cooling observed between April and July is due to the decrease of solar radiation probably due to cloud cover, with added effect of zonal heat advection. We observe the warmest temperature in March and the minimum is reached in August, which corresponds to the upwelling season (Ostrowski et al., 2009; Kopte et al., 2017).

Although the contribution of zonal heat advection remains weak compared to the solar flux, the variations of zonal heat advection are in phase with the variation of heat tendency throughout the year.

In the Benguela region, the heat tendency variations are roughly in phase with the heat flux variations. The heat budget is mostly driven by the shortwave flux, as found previously in the neighboring southern Angola upwelling system. The cooling that occurs from March to August can be associated with cloud cover, which reduces the incoming solar flux, and also a small

contribution of oceanic processes. The observed coldest temperatures correspond to the July-October upwelling season (Hagen et al., 2001; Muller et al., 2014).

The Senegal region is the only one among the four analyzed regions where the salt budget is clearly controlled by the surface freshwater fluxes, with added runoff effect (Fig. 8b). From March to October the observed freshening can be explained by the

combined effect of precipitation, and Senegal and Gambia rivers inputs. From October to November, zonal salt advection adds its contribution to existing freshwater inputs to freshen the mixed-layer. Vertical salt diffusion, with additional contribution of meridional and vertical salt advection, tends to increase mixed-layer salinity and partly compensates the previous freshening effect. Although our selected regions are slightly different, these results are until then consistent with those of Camara et al. ( 2015). However, in our study, evaporation plays a dominant role to increase salinity for the rest of year, even if there is also a

weak contribution of oceanic processes. This disagrees with the study of Camara et al. (2015), where the contribution of evaporation to the mixed-layer salt budget is very weak compared to our results. This contradiction can be explained by different model configurations as described in (Da-Allada et al., 2017). Camara et al. (2015) use for model forcing an older version of the Drakkar forcing set (DFS4) compared to the one (DFS5.2) used in the present study, and their model has less vertical levels than ours (46 vs 75). They also use a smaller density criterion (0.01 kg m$^{-3}$ for Camara et al. (2015) vs 0.03 kg

m$^{-3}$ in this study to estimate mixed-layer depth), and an additional restoring term for salinity in their model.



In the Equatorial region, the seasonal variability of mixed-layer salinity is mainly due to oceanic processes as shown in other studies (Da-Allada et al., 2013, 2014). In our study, we found vertical salt diffusion and zonal salt advection as dominant oceanic processes (Fig. 8f). From October to December and March to July, zonal advection is the most important freshening contribution, stronger that precipitation. It is explained by the westward South Equatorial Current (SEC), which transports the low salinity waters from the Gulf of Guinea associated with the Niger and Congo River plumes (Houndegnonto et al., 2021). The major role played by vertical salt diffusion to increase the salinity in the mixed-layer, demonstrated in previous studies (Da-Allada et al., 2014, 2017), is confirmed by our results for boreal spring/summer in particular. This strong vertical salt diffusion is the consequence of the vertical shear between the westward SEC and the eastward EUC (which transports high salinity waters), but can be reduced by the strong salinity stratification caused by Niger and Congo river plumes (Jouanno et al., 2011). Note that vertical diffusion is however strongly compensated by zonal advection, above all in May. These results agree with the other studies covering the region despite slight differences in the limits of selected boxes (Berger et al., 2014; Da-Allada et al., 2014; Camara et al., 2015; Da-Allada et al., 2017). Although the contribution of surface freshwater fluxes and runoff remains weak in our study, its seasonal variations follow those described by (Da-Allada et al., 2017), with some time lag. During the period when the ITCZ is close to the equator between November and April, the freshwater flux is dominated by precipitation and decreases the mixed-layer salinity, whereas the rest of year, the freshwater flux is dominated by evaporation and increases salinity.

As in the Equatorial region, horizontal and vertical oceanic processes drive the mixed-layer salinity in the Angola region too (Fig. 8g-i), in agreement with previous studies (Camara et al., 2015; Awo et al., 2022). Meridional salt advection explains most of the variability in salt budget, and particularly its semi-annual cycle, as it freshens the mixed-layer in February-April and September-October, when the southward Angola Current brings low-salinity water from the Congo River plume (Gordon and Bosley, 1991; Awo et al., 2022). However, for the rest of the seasonal cycle, a combined action of meridional salt advection, vertical salt advection and vertical salt diffusion increases the salinity of mixed-layer. Although vertical advection is stronger than vertical diffusion, they remain in phase throughout the cycle and act against the runoffs and zonal salt advection. Seasonal variations in both vertical salt diffusion and advection are driven by changes in the vertical salinity gradient related to the semi-annual intrusion of low salinity surface waters (Camara et al., 2015; Awo et al., 2022).

In the Benguela region, individual contributions of physical processes are relatively weak in compared to the other regions. The mixed-layer salinity variability is partly controlled by freshwater fluxes and particularly evaporation (Fig. 8j-k). Zonal advection remains negative throughout the year, and from March to December, it acts to decrease salinity, against the action of evaporation that is reinforced by vertical salt diffusion between September and December. The increase of salinity corresponds to the upwelling season in the southern part of Benguela upwelling system in summer (Muller et al., 2014).

Although increasing resolution in oceanic models intends to produce more realistic simulations by explicitly resolving mesoscale variability, and models are the only way to estimate all terms of the heat/salt budget in the mixed-layer, it is difficult to directly validate such model budgets with in situ data. One problem is that globally available in situ data can only explicitly





resolve near-surface horizontal processes and particularly advection, not vertical processes that have to be estimated as a residual. A second problem is that in situ observation density does not allow estimating horizontal advection at the high resolution available from models. Therefore, to be properly compared with those available from observations, model horizontal advection terms must be computed offline at the spatio-temporal resolution of observations. Our results indeed show that the time-averaged spatial distribution of NEMO offline heat/salt advection terms compares much better to PREFCLIM horizontal

advection terms than the online heat/salt advection terms. However, when examining the seasonal cycle of horizontal advection in selected boxes, NEMO offline terms do not always compare well with PREFCLIM, sometimes less than online terms. This suggests that temporal coverage of in situ observations is more critical that spatial coverage, particularly for salinity, and especially in coastal areas of Africa where Argo profiles are relatively scarce and equatorial region where Lagrangian drifters do not stay long due to Ekman divergence. Another possibility would be to estimate advection from satellite products of SST,

SSS and currents, the latter estimated from altimetry and satellite wind for their geostrophic and Ekman components respectively (Bonjean and Lagerloef, 2002), which are available at a resolution of a few tens of kilometers and a few days. The new SWOT mission (Morrow et al., 2019) should soon improve further the resolution of geostrophic currents. The often large differences between offline and online advection terms in the model suggest an important role of small scale ($< 2.5°$, $<$ 1 month) variability, which includes mesoscale activity recently documented in the eastern tropical Atlantic (Aguedjou et al.,

2019), including at the equator where differences are particularly large. Although the model mixed-layer budgets validation has some limitations in the present study, our results are generally in agreement with earlier studies of mixed-layer heat/salt budgets in Atlantic tropical (Foltz et al., 2003; Jouanno et al., 2011; Wade et al., 2011; Da-Allada et al., 2013; Camara et al., 2015). Except for local studies where coordinated field measurements and mooring deployment can be combined to close a short-term mixed-layer budget with in situ observations (Farrar et al., 2015; Farrar and Plueddemann, 2019), in most regions,

one can only use models to close mixed-layer budgets and thrust them to quantify the processes hidden in the residual unresolved by global observations.







## Appendix A

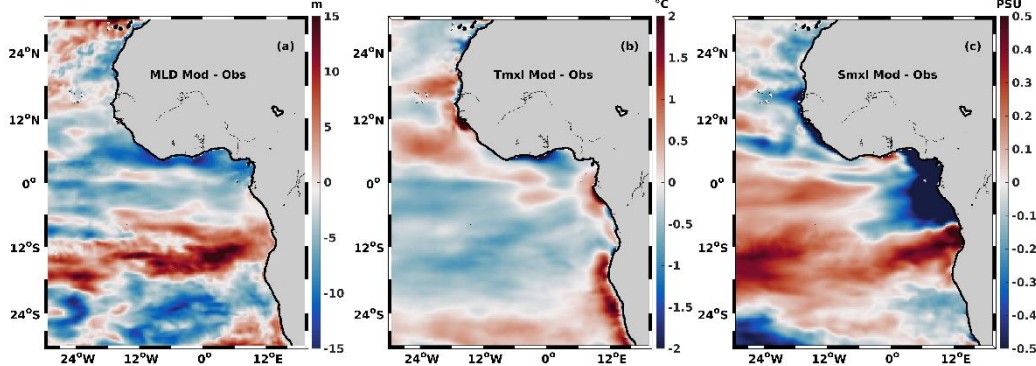

**Figure A.1:** Difference between observations and model in mixed-layer depth (a), mixed-layer temperature (b) and mixed-layer salinity (c).

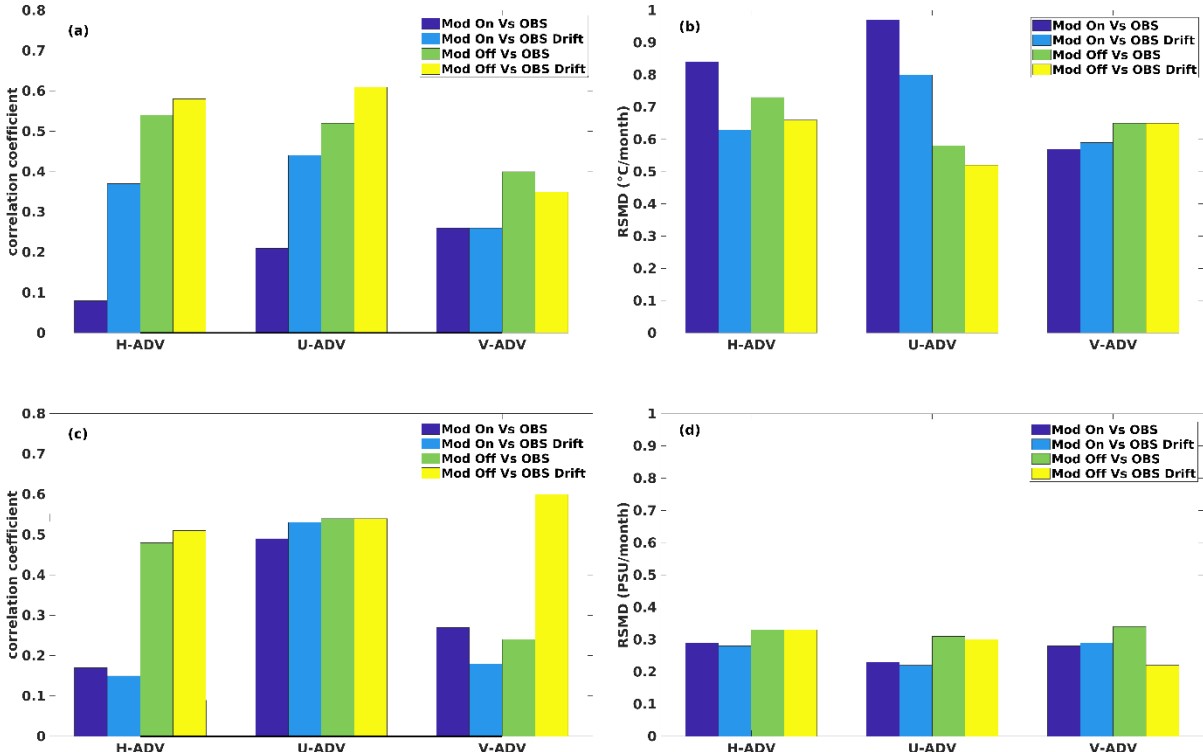

**Figure A.2:** Spatial correlation (r), RMSD of horizontal, zonal and meridional heat (a and b)/salt (c and d) advection between observations (Obs and Obs Drift) and model (online and offline).



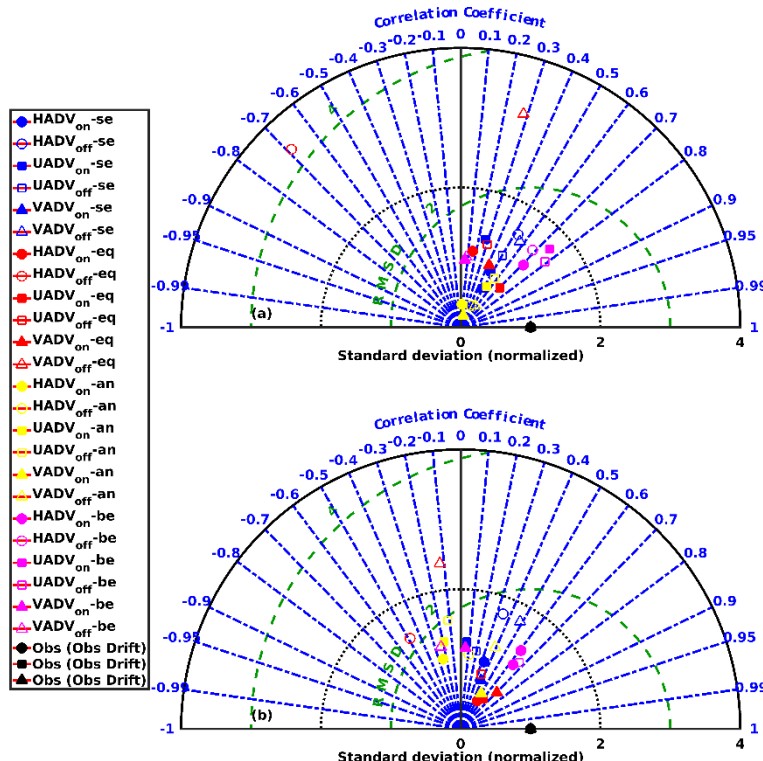

**Figure A.3:** Taylor diagrams comparing seasonal variations of horizontal/zonal/meridional heat advection (HADV/UADV/VADV) from observations (gridded advection named Obs (a) ; Lagrangian advection named Obs Drift (b)) and model (online advection for ON and offline advection for OFF) in Senegal, Equatorial, Angola and Benguela boxes (blue, red, yellow and magenta colors respectively).




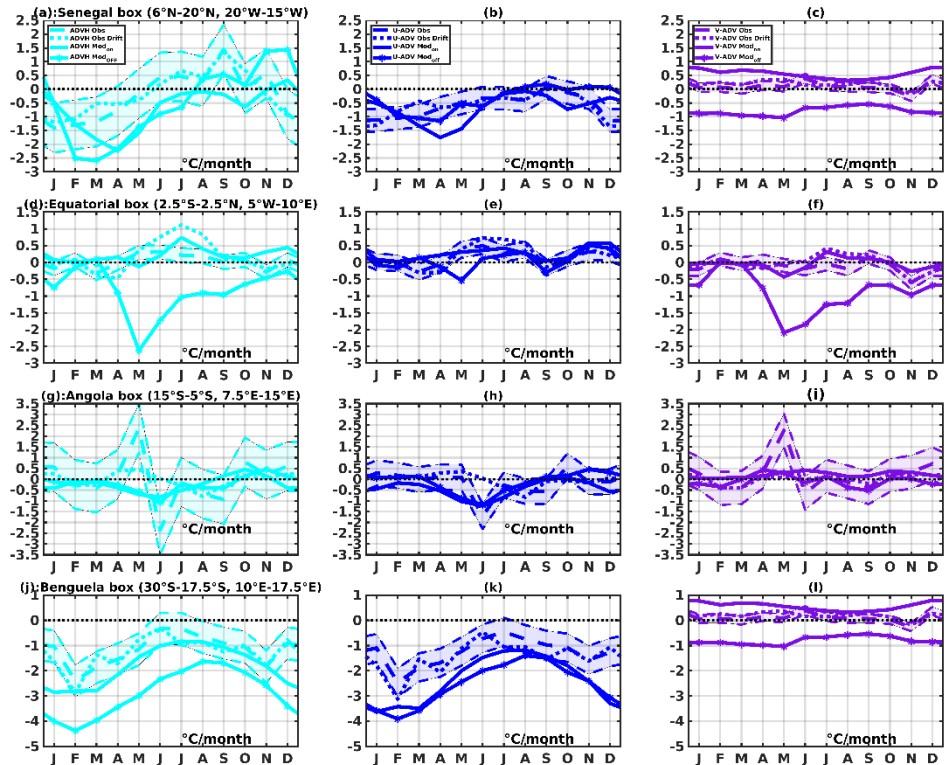

**Figure A.4:** Seasonal cycle of horizontal (cyan), zonal (blue), and meridional (purple) heat advection from observations (dashed line for gridded advection and dotted line for Lagrangian advection) and model (full line for online and full dotted line for offline) in selected boxes. All others terms are in °C per month.




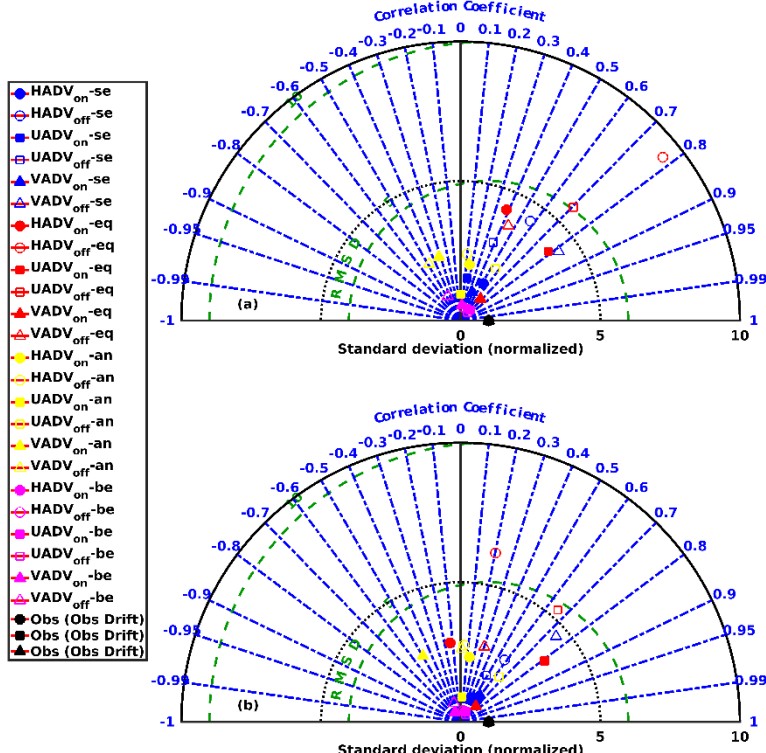

**Figure A.5:** Taylor diagram of horizontal salt advection and these components between observations (gridded advection for Obs (a) and Lagrangian advection for Obs Drift (b)) and model (online for ON and offline for OFF). Senegal, Equatorial, Angola and Benguela boxes are designed by blue, red, green and magenta colors respectively.




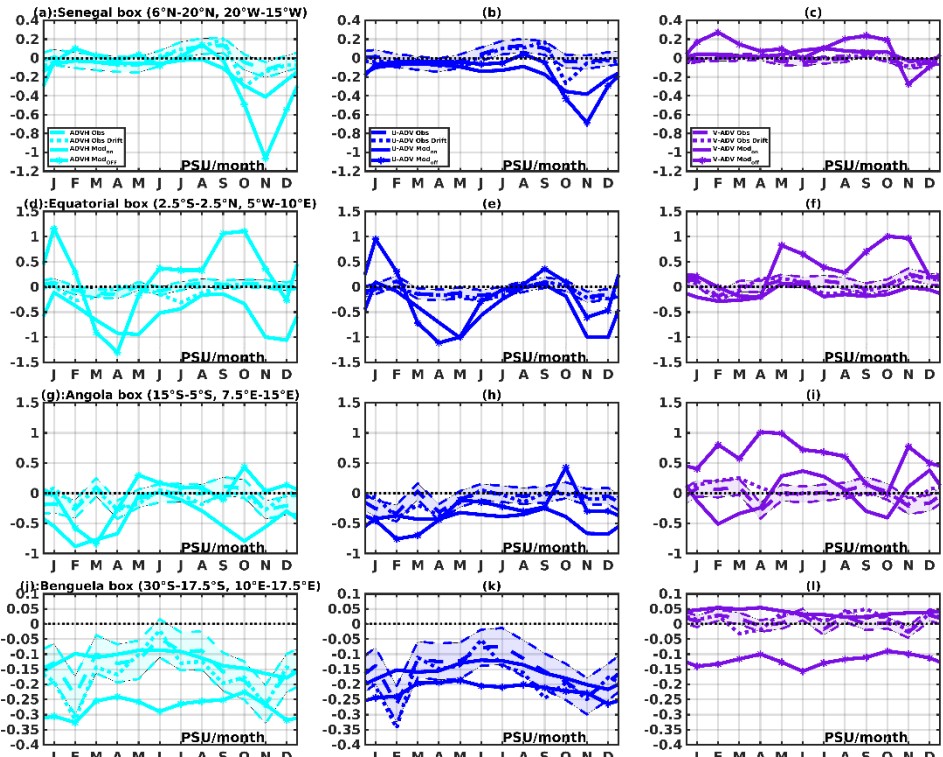

**Figure A.6:** Seasonal cycle of horizontal (cyan), zonal (blue), and meridional (purple) salt advection from observations (dashed line for gridded advection and dotted line for Lagrangian advection) and model (full line for online and full dotted line for offline) in selected boxes. All others terms are in Psu per month.

## Data availability

The PREFCLIM climatology used here is available from https://doi.pangaea.de/10.1594/PANGAEA.868927. Model simulations are available from authors on demand.

## Author contribution

RN performed the data analysis and wrote the manuscript with strong contribution from GA. OK did some preliminary analysis under supervision of GA, CD-A and JJ. WR and JJ produced the PREFCLIM climatology and the NEMO model simulation respectively. All co-authors contributed to the scientific improvement of the manuscript.

## Competing interests

The authors declare that is not an interest conflict about this performed research.





**Acknowledgements**

This work is part of the PhD thesis of R. Ngakala, funded by DAAD (Deutscher Akademischer Austauschdienst / German Academic Exchange Service) in the frame of the "In-Country/In-Region Scholarship Programme" Sub-Saharan Africa. The PREFCLIM climatology was produced in the frame of the European Union FP7 PREFACE project. This study was supported by the TRIATLAS project, which has received funding from the European Union's Horizon 2020 research and innovation program under grant agreement 817578. This study is also supported by the TOSCA SMOS and SWOT-GG projects funded
by CNES.

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
