# Peer review of "Joint observation-model mixed-layer heat and salt budgets in the eastern tropical Atlantic"

_EGUsphere, 2022_

## Referee Comment (RC2)

General comments

This study is focusing on the understanding of the key process that drive the mean seasonal cycle of heat and salt in the tropical Atlantic. They use data from model (NEMO) and observation (PREFCLIM) and mixed layer budget technique for this purpose. The paper is well written and the flow is to follow.

While the topic is interesting, there are issues the authors must address before publications (detail below). Therefore, I recommend minor revision.

Minor comments

1°) I don't understand what the added value of this paper is. The authors should emphasize this point. To me, most of the results have already been shown previously. The originality of this paper should be clearly addressed.

2°) Why don't you forced the model with the same heat flux as in PFREFCLIM (or vice-versa) since it allows to do that. This will allow, at least, to overcome the problem found in the inconsistency of heat flux in the interpretation of results.

3°) The seasonal variations of the salinity tendency are quite different from Obs and Model. Can you explain this?

5°) Is the use of PREFCLIM suitable for this kind of study since it provides a coarse temporal and spatial resolution?

3°) What is the criterion used to delimitate the boxes? You should clarify this in the text.

Line 37: are you sure that paper of Giordani and Caniaux, 2001 deals about the altantic cold tongue ?

Line 257:  Please can you explain why there a discrepancy between the model and the Obs only at the equator ?

---

## Author Comment (AC2)

EGUsphere, referee comment RC1
https://doi.org/10.5194/egusphere-2022-1245-RC1, 2022 © Author(s)
2022. This work is distributed under the Creative Commons Attribution
4.0 License.

**Comment on egusphere-2022-1245**

Anonymous Referee #1

Referee comment on "Joint observation-model mixed-layer heat and salt budgets in the eastern tropical Atlantic" by Roy Dorgeless Ngakala et al., EGUsphere, https://doi.org/10.5194/egusphere-2022-1245-RC1, 2022

We thank the anonymous reviewer #1 for these constructive comments to improve our manuscript. We also appreciate the recommended papers on the importance of tropical instability waves (TIWs) in the eastern equatorial Atlantic heat budget, which we found useful. Below, we have tried to provide responses (blue text) to the reviewer's comments (black text), and we hope that the revised manuscript will satisfy the reviewer.

My most serious concern is that there is no discussion of the importance of tropical instability waves in the heat balance of the eastern equatorial Atlantic. The effect of TIWs is hidden in the lateral diffusion term (D-sub-L) as well as in the vertical turbulent diffusion term (because of the effects of TIWs on vertical shear, e.g. Heukamp et al., 2022). These important processes are not discussed at all and yet there is a long history of describing their role in the near equatorial heat balance (e.g., Weisberg and Weingartner, 1988; Grodsky et al, 2005; Lee et al, 2014; and many more). I'm guessing the authors have ignored this issue because TIWs have periods of ~30 days whereas PREFCLIM is a monthly climatology (although its temporal resolution is never specified-see below). Thus, PREFCLIM does not resolve TIWs. However, the model has no such limitation in terms of temporal resolution. The model is used to assess what is lost in terms of the effects of sub-mesoscale variability in computing the surface layer heat balance from PREFCLIM. The model can be used in a very similar way to assess the effects of mesoscale TIWs that are not resolved by PREFCLIM.

Response 1:

We agree on the importance of tropical instability waves (TIWs) in the heat budget in the equatorial Atlantic region, although it was not addressed in the submitted manuscript. As mentioned by the reviewer, there are many studies, either based on observations (Grodsky et al., 2005; Lee et al., 2014; Heukamp et al., 2022) or models (Jochum et al., 2004, 2005; Peter et al., 2006) , describing the role of TIWs in the tropical Atlantic. In the equatorial Atlantic, the TIWs are mostly found slightly north of the equator, outside of the Gulf of Guinea between 30°W and 10°W (Olivier et al., 2020; Tuchen et al., 2022). The role often attributed to TIWs in the mixed-layer heat budget is warming by horizontal advection and cooling by vertical mixing of the equatorial cold tongue (Jochum et al., 2004; Grodsky et al., 2005; Peter et al., 2006). On the one hand, horizontal advection due to TIWs will bring warm water from the eastern basin across the South Equatorial Current (SEC), thus warm the cold tongue water (Foltz, 2003; Peter et al., 2006; Tuchen et al., 2022). On the other hand, TIWs will amplify the vertical shear between the equatorial current system, thus enhance the vertical mixing between the warm surface water and the cold subsurface water, and consequently enhancing the surface cooling (Jochum et al., 2005; Heukamp et al., 2022). In our study, we did not focused on TIWs and, unfortunately, could not isolate advection at specific timescales due to the unavailability of daily simulation outputs. Indeed, we exploit here monthly outputs from a simulation produced a few years ago, do not have computational time available at the moment in our laboratory to rerun the simulation to produce daily outputs, and wonder if it would be worth the carbon cost. Nonetheless, our subsampling method allows to isolate submesocale variability. As mentioned by the reviewer, the effect of TIWs can be hidden in the lateral diffusion term computed from models. However, comparing two simulations at low (1° spatial resolution)

and high (0.25° spatial resolution) resolutions to study the effect of TIWs in the heat budget, Jochum et al. (2005) showed that, at low resolution, the effect of TIWs is included in the strong lateral diffusion term, while at high resolution, the effect of TIWs is included in the heat advection term and the lateral diffusion term is much weaker. Indeed, in our high-resolution (0.25° like Jochum et al. 2005) simulation, the lateral diffusion term is generally much weaker than the horizontal advection term in the mixed-layer heat budget (compare Figure R1.1a and Figure R1.2.a below), including in the TIW region. Also the positive difference between online and offline advection, particularly strong in the TIW region (Figure R.1.2), strongly supports the idea that the TIWs contribution is transferred from the horizontal diffusion term to the horizontal advection term when increasing resolution, which more generally allows to explicit resolve smaller scales. TIWs also contribute to vertical diffusion, as underlined by the reviewer, which explain why this process is particularly strong in the TIWs region (Figure R1.1b), but it is still strong farther east in the equatorial part of the Gulf of Guinea due to the current shear associated with the EUC (Jouanno et al., 2011). Although we do not expect any significant contribution of TIWs in our boxes where the seasonal budgets are presented, as these boxes are outside of the TIWs region, we now added in the paper some discussion about the role of TIWs when presenting regional maps including the TIW region and discussing the processes involved in our equatorial box (see lines 106-113 "page 4", lines 532-538 "pages 29-30", lines 562-564 "page 30", and line 567-568 "page 30" in revised manuscript).

[Figure]

Figure R1.1: Mean lateral (a) and vertical (b) heat diffusions from NEMO model at original 0.25° resolution. The black rectangles correspond to our 4 regions of study. The red rectangle represents the TIWs box as used by Tuchen et al. (2022).

[Figure]

Figure R1.2: Mean horizontal heat advection from model: online re-sampled at 2.5° resolution (a), offline (b) and their difference (c). The black rectangles correspond to the 4 regions of study. The red rectangle represents the TIWs box as used in Tuchen et al. (2022).

Introduction. Somewhere around lines 50-70, the paper by Scannell and McPhaden (2018) should be cited and discussed.

Response 2:

As recommended by the reviewer, we have introduced this article from line 69 in the revised manuscript as follows:

 **"The dominance of the surface heat flux is highlighted by a recent study based on PIRATA buoy data off the equator, in particular at the 6°S,8°E position. At this latitude, seasonal variations in SST governed by solar flux and latent heat flux are shown to be associated with the meridional migration of the ITCZ and the formation of low-level marine stratocumulus , (Scannell and McPhaden 2018) ."** (See lines 69-72 "page 3").

And we briefly discussed this , Scannell and McPhaden (2018) paper from line 552 as follows:

**"Recently, Scannell and McPhaden (2018) also confirmed the role of turbulent vertical mixing from a PIRATA buoy located at the southeastern edge of the ACT."** (See lines 552-553 "page 30").

 And from line 571 as follows:

**"Competition between the shortwave flux and the latent heat flux is also mentioned in (Scannell and McPhaden, 2018). Although at the 6°S,8°E position, the horizontal advection remains weak in their study**. (See Lines 571-572 "page 31")"

Line 85. The PIRATA acronym should be defined and a reference provided (e.g. Bourles et al, 2019).

response 3:

The reference (Bourlès et al., 2019) is added after the acronym PIRATA in the revised version (See line 135 "page 5").

Section 2.1.1. The temporal resolution of PREFCLIM is not specified. What is it?

response 4:

The temporal resolution (monthly) of PREFCLIM is now specified in the Observations subsection (See line 131 "page 5").

Figure 3. The Angola and Equatorial boxes contain areas of very high variability and very low variability. Presumably, the areal averages are therefore representative of much smaller regions within the boxes where the variability is high. This bias should be noted and discussed.

Response 5:

We thank the reviewer for this remark and did sensitivity tests on the boxes, by decomposing each one into two boxes (figures R1.3 & R1.4 below). For the equatorial box, except for the observed shift in the salinity tendency in March in the western half (2.5°S-2.5°N, 5°W-2°E) compared to the eastern half (2.5°S-2.5°N, 2°E-10°E) and the full box (2.5°S-2.5°N, 5°W-10°E), we note the same seasonal variations in SSS, related salinity tendency, and associated dominant processes. However, the magnitude of the individual processes increase toward the east with the salinity stratification of due to the contribution of the rivers. After decomposing the Angola box (15°S-5°S, 7.5°E-15°E) into its northern half (10°S-5°S, 7.5°E-15°E) and southern half (15°S-10°S, 7.5°E-15°E), we observe that the seasonal variations in the mixed-layer salt budget are all nearly identical but with different intensity for each process involved. Due to this relatively small sensitivity of the salt budget to the box boundaries, although the Angola and Equator boxes consist of areas of high and low variability in SSS, we decided to keep these boxes for consistency between mixed-layer heat/salt budgets. The selection of the boxes, which cover regions with particularly low mean SST/SSS and/or strong SST/SSS variability, is necessarily a trade-off. We therefore noted and discussed this bias as follows in the revised version (see lines 265-270 "page 12", lines 455-457 "page 24", and lines 474-476 "page 24").

[Figure]

Figure R1.3: Seasonal mixed-layer salt budget terms from observations (dashed line) and model (full line and full dotted line for pseudo-residual associated with offline advection) in the equatorial box used in the paper (top row), its eastern half (middle row) and western half (bottom row). SSS in PSU and tendency terms in PSU per month.

[Figure]

Figure R1.4: Seasonal mixed-layer salt budget terms from observations (dashed line) and model (full line and full dotted line for pseudo-residual associated with offline advection) in the Angola box used in the paper (top row), its northern half (middle row) and southern half (bottom row). SSS in PSU and tendency terms in PSU per month.

Line 275. What are the implications of the off-line advection correlating better with The Lagrangian estimate of advection? Similar question regarding statement in lines 298-99 comparing offline/online to PREFCLIM.

Response 6:

Lagrangian advection terms in PREFCLIM were obtained from the estimated velocities at each point along the drifters and floats trajectories combined with the corresponding high-resolution hydrographic climatology, then gridded. We expected that online advection could compare better to Lagrangian advection than offline advection due to the finer resolution relative to the float trajectories. However, we get the opposite result. This is probably explained by the fact that float trajectories are not homogeneous, and by the low spatial coverage of floats in the Gulf of Guinea, as now noted (see lines 300-301 "page 13"). In addition, we could also expect the offline advection to compare better with PREFCLIM "Gridded advection" than the online advection at the seasonal time scale, as it was the case for the spatial comparison (Fig. 4e-g), again showing that non-linear terms cannot be properly captured by observations. However, note that this is not verified in all cases, and this disagreement can probably be associated with the temporal coverage of the in situ observations as discussed in the last section of the manuscript (see lines 650-653 "page 33").

Figures 5 and 8. I do not see full dotted lines in these figures.

Response 7:

Improved versions of these figure are now provided.

The Rath and Dengler (2016) reference seems incomplete.

Response 8:

Rath et al. (2016) is in fact complete, as can be checked by following the doi link: https://doi.pangaea.de/10.1594/PANGAEA.868927. This is a dataset, which explains why there is no journal title.

References

Bourlès, B., Araujo, M., McPhaden, M. J., Brandt, P., Foltz, G. R., Lumpkin, R., Giordani, H., Hernandez, F., Lefèvre, N., Nobre, P., Campos, E., Saravanan, R., Trotte-Duhà, J., Dengler, M., Hahn, J., Hummels, R., Lübbecke, J. F., Rouault, M., Cotrim, L., Sutton, A., Jochum, M., and Perez, R. C.: PIRATA: A Sustained Observing System for Tropical Atlantic Climate Research and Forecasting, Earth Sp. Sci., 6, 577–616, https://doi.org/10.1029/2018EA000428, 2019.

Foltz, G. R.: Seasonal mixed layer heat budget of the tropical Atlantic Ocean, J. Geophys. Res., 108, 1–13, https://doi.org/10.1029/2002jc001584, 2003.

Grodsky, S. A., Carton, J. A., Provost, C., Servain, J., Lorenzzetti, J. A., and Mcphaden, M. J.: Tropical instability waves at 0 ° N , 23 ° W in the Atlantic : A case study using Pilot Research Moored Array in the Tropical Atlantic ( PIRATA ) mooring data, 110, 1–12, https://doi.org/10.1029/2005JC002941, 2005.

Heukamp, F. O., Brandt, P., Dengler, M., Tuchen, F. P., McPhaden, M. J., and Moum, J. N.: Tropical Instability Waves and Wind-Forced Cross-Equatorial Flow in the Central Atlantic Ocean, Geophys. Res. Lett., 49, e2022GL099325, 2022.

Jochum, M., Malanotte-Rizzoli, P., and Busalacchi, A.: Tropical instability waves in the Atlantic Ocean, Ocean Model., 7, 145–163, https://doi.org/10.1016/S1463-5003(03)00042-8, 2004.

Jochum, M., Murtugudde, R., Ferrari, R., and Malanotte-Rizzoli, P.: The Impact of Horizontal Resolution on the Tropical Heat Budget in an Atlantic Ocean Model, J. Clim., 18, 841–851, https://doi.org/10.1175/JCLI-3288.1, 2005.

Lee, T., Lagerloef, G., Kao, H.-Y., McPhaden, M. J., Willis, J., and Gierach, M. M.: The influence of salinity on tropical Atlantic instability waves, J. Geophys. Res. Ocean., 119, 8375–8394, https://doi.org/10.1002/2014JC010100, 2014.

Olivier, L., Reverdin, G., Hasson, A., and Boutin, J.: Tropical Instability Waves in the Atlantic Ocean: Investigating the Relative Role of Sea Surface Salinity and Temperature From 2010 to 2018, J. Geophys. Res. Ocean., 125, 1–17, https://doi.org/10.1029/2020JC016641, 2020.

Peter, A. C., Le Hénaff, M., du Penhoat, Y., Menkes, C. E., Marin, F., Vialard, J., Caniaux, G., and Lazar, A.: A model study of the seasonal mixed layer heat budget in the equatorial Atlantic, J. Geophys. Res. Ocean., 111, 1–16, https://doi.org/10.1029/2005JC003157, 2006.

Rath, W., Dengler, M., Lüdke, J., Schmidtko, S., Schlundt, M., Brandt, P., Bumke, K., Ostrowski, M., van der Plas, A., Junker, T., Mohrholz, V., Sarre, A., Tchipalanga, P. C. M., and Coelho, P.: PREFCLIM: A high-resolution mixed-layer climatology of the eastern tropical Atlantic, https://doi.org/10.1594/PANGAEA.868927, 2016.

Scannell, H. A. and McPhaden, M. J.: Seasonal Mixed Layer Temperature Balance in the Southeastern Tropical Atlantic, J. Geophys. Res. Ocean., 123, 5557–5570, https://doi.org/10.1029/2018JC014099, 2018.

Tuchen, F. P., Perez, R. C., Foltz, G. R., Brandt, P., and Lumpkin, R.: Multidecadal Intensification of Atlantic Tropical Instability Waves, Geophys. Res. Lett., 1–11, https://doi.org/10.1029/2022gl101073, 2022.

Weisberg, R. H. and Weingartner, T. J.: Instability Waves in the Equatorial Atlantic Ocean, J. Phys. Oceanogr., 18, 1641–1657, https://doi.org/10.1175/1520-0485(1988)018<1641:IWITEA>2.0.CO;2, 1988.

---

## Author Comment (AC3)

We thank anonymous reviewer #2 for his availability to review our manuscript and his suggestions will be taken into account. In what follows, our responses and the reviewer's comments are represented by blue and black text, respectively.

1°) I don't understand what the added value of this paper is. The authors should emphasize this point. To me, most of the results have already been shown previously. The originality of this paper should be clearly addressed.

Response 1:

It is true that our results on driving processes of mixed layer heat and salt budgets in the tropical Atlantic are mostly consistent with previous results, but we believe that the originality of our study lies in several other points:
1- While the PREFCLIM mixed layer climatology was previously published as a dataset (Dengler and Rath, 2015; Rath et al., 2016), it is the first study that scientifically exploits this dataset.
2- To our knowledge, heat/salt advective terms have been previously estimated by model, observations or both (reanalyses) but independent estimations from model on one side, and from observations on the other side, have rarely been confronted (if ever).
3- The important role of mesoscale variability (and the subsequent need for subsampling to compare modelled and observed advection) had not been previously quantified in the region of study, and this was done here in the frame of projects related to new/future satellite missions (SWOT, SMOS-HR) which should help to capture this mesoscale variability.
While we believe point 3 was already clearly stated (see last paragraph of part 4). The first 2 points are now strengthened in the introduction (see lines 115-118 "page 4").

2°) Why don't you forced the model with the same heat flux as in PFREFCLIM (or vice-versa) since it allows to do that. This will allow, at least, to overcome the problem found in the inconsistency of heat flux in the interpretation of results.

Response 2:

This difference in the heat flux comes from the fact that the PREFCLIM climatology and the NEMO tropical Atlantic simulation compared here were independently produced by two different research groups (German GEOMAR for the PREFCLIM climatology and French LEGOS for the NEMO simulations), who collaborate in this paper. We appreciate the suggestion of the reviewer to overcome the inconsistency problem of the heat flux. But unfortunately it is not so easy to update the PREFCLIM product or NEMO simulation. In particular we do not have computational time available at the moment in our laboratory to rerun the simulation with different heat forcing fluxes, and wonder if it would be worth the carbon cost. Actually, only the solar flux (SWR) is used for forcing the model, while non-solar fluxes are computed through bulk formulas, allowing feedback from simulated oceanic conditions (this now clarified at lines 285-286 "page 13" in the paper), and differences between NEMO and PREFCLIM are larger for the latent heat flux (LHF) than for the SWR (see figure R2.1). Also, the heat flux term for PREFCLIM and NEMO depends also on the mixed layer of each product. Comparing Figs R2.1 below and Fig. 4a-b, it appears that, although SWR is stronger in the model (particularly in the south-west corner), the net heat flux is generally weaker in the model, so the different solar fluxes used for NEMO and PREFCLIM has no direct influence. Moreover, the differences in solar flux forcing does not affect the differences in advective terms which are the main focus in the paper.

[Figure]

Figure R2.1: Difference of components of heat flux between model and observations: short wave radiation (a), latent heat flux (b), long wave radiation (c) and sensible heat flux (d). Except of SWR, the difference of other is computed from absolute values: abs(model) - abs(observation).

3°) The seasonal variations of the salinity tendency are quite different from Obs and Model. Can you explain this?

Response 3:

The differences in the seasonal variations of the salinity tendency between Obs and Model directly derive from the differences in the mixed-layer salinity seasonal variations (Fig. 8a/d/g/j), which have generally a larger amplitude and lower minima in the model, as also seen in Fig. 3. This can be due to several causes. First, although the PREFCLIM product benefited from newly available hydrographic data in the Senegal, Angola and Namibia coastal waters, the data density is still low in the equatorial Gulf of Guinea (Dengler and Rath, 2015), where lie the freshest waters associated with heavy rains and the large Congo and Niger river plumes, hence the largest difference in seasonal salinity variations in the Equatorial box (Fig. 8d). Poor data density can also be associated with a seasonal bias that may prevent capturing the full seasonal cycle. Second, hydrographic profiles, notably those from Argo floats, do not sample the salinity minimum found in the upper few meters of the ocean in regions highly stratified by rain and river plumes, which induces SSS estimations higher than those observed from satellite (Boutin et al., 2016; Houndegnonto et al., 2021). This leads to overestimation of mixed-layer salinity too. Third, while the NEMO model configuration has high vertical resolution in the upper few meters and homogeneous spatial coverage, the way it reproduces mixed-layer salinity highly depends on its freshwater forcing, including river runoff, and its own dynamics that are of course not perfect. This discussion has been added in the paper (see lines 584-596 "page 31").

4°) Is the use of PREFCLIM suitable for this kind of study since it provides a coarse temporal and spatial resolution?

Response 4:

Yes, we believe it was worth using PREFCLIM in this study. Indeed, the coarse temporal and spatial resolution of the PREFCLIM climatology (for heat/salt budget terms) does not allow the consideration of nonlinear terms related to mesoscale activity. This problem could be solved with the arrival of new satellite missions for high frequency variability such as SWOT, SMOS-HR, as noted in the paper (see lines 656-658 "page 33"). But the comparison of PREFCLIM with offline NEMO also shows that there is still uncertainty in other budget terms, and it is always important to have some observations to validate a model.

5°) What is the criterion used to delimitate the boxes? You should clarify this in the text.

Response 5:

The criteria are partly subjective, but we now tried to precise them, and also note that there is a trade-off between SST and SSS characteristics (see lines 262-270 "page 12").

Line 37: are you sure that paper of Giordani and Caniaux, 2001 deals about the atlantic cold tongue?

Response 6:

Thanks for pointing at that. It was a mistake, it has been replaced by Caniaux et al. (2011) in the revised manuscript.

Line 257: Please can you explain why there a discrepancy between the model and the Obs only at the equator?

Response 7:

As pointed out above, this disagreement is partly associated with the different data sources used for heat fluxes in the observations (TropFlux) and as forcing in the model (DFS5.2). However, there is in places more disagreement in non-solar fluxes calculated by the model through bulk formulas (see figure R2.2). These causes for the differences are cited in the paper (see lines 284-286 "page 13"). A sensitivity test based on the weak correlations (see Figure 4.d of the manuscript) between heat fluxes from observations and the model west of 12°W in the equatorial band (3°S-3°N, West of 12°W) shows most important differences in the solar short-wave flux and latent heat flux between observations and the model. We think it is beyond the scope of the paper to investigate further reasons for these differences.

[Figure]

Figure R2.2: Mean of different components of heat flux from observations (a,d,g,j) and model re-sampled at 2.5° resolution (b,e,h,k): short wave radiation (a,d), long wave radiation (d,e), latent heat flux (g,h) and sensible heat flux (j,k). (c,f,i,l) represent the seasonal correlation of these components between observations and model, which is 95% significant when r > 0.58. Over (b,e,h,k), r indicates the spatial correlation between observations and model, which is 95% significant when r > 0.12.

[Figure]

Figure R2.3: Seasonal cycle of heat flux and this decomposition from observations (dashed line) and model (full line) in selected region (3°S-3°N, West of 12°W for sensitivity test). All terms are in °C per month. r indicates the seasonal correlation between observations and model, which is 95% significant when r > 0.58.

Reference

Boutin, J., Chao, Y., Asher, W. E., Delcroix, T., Drucker, R., Drushka, K., Kolodziejczyk, N., Lee, T., Reul, N., Reverdin, G., Schanze, J., Soloviev, A., Yu, L., Anderson, J., Brucker, L., Dinnat, E., Santos-Garcia, A., Jones, W. L., Maes, C., Meissner, T., Tang, W., Vinogradova, N., and Ward, B.: Satellite and in situ salinity understanding near-surface stratification and subfootprint variability, Bull. Am. Meteorol. Soc., 97, 1391–1407, https://doi.org/10.1175/BAMS-D-15-00032.1, 2016.

Dengler, M. and Rath, W.: Seasonal heat and fresh water mixed-layer balance climatology, 1–15 pp., 2015.

Houndegnonto, O. J., Kolodziejczyk, N., Maes, C., Bourlès, B., Da-Allada, C. Y., and Reul, N.: Seasonal Variability of Freshwater Plumes in the Eastern Gulf of Guinea as Inferred From Satellite Measurements, J. Geophys. Res. Ocean., 126, 1–27, https://doi.org/10.1029/2020JC017041, 2021.

Rath, W., Dengler, M., Lüdke, J., Schmidtko, S., Schlundt, M., Brandt, P., Bumke, K., Ostrowski, M., van der Plas, A., Junker, T., Mohrholz, V., Sarre, A., Tchipalanga, P. C. M., and Coelho, P.: PREFCLIM: A high-resolution mixed-layer climatology of the eastern tropical Atlantic, https://doi.org/10.1594/PANGAEA.868927, 2016.

---

## Author Response (AR2)

EGUsphere, referee comment RC1
https://doi.org/10.5194/egusphere-2022-1245-RC1, 2022 © Author(s)
2022. This work is distributed under the Creative Commons Attribution
4.0 License.

**Comment on egusphere-2022-1245**

Anonymous Referee #1

Referee comment on "Joint observation-model mixed-layer heat and salt budgets in the eastern tropical Atlantic" by Roy Dorgeless Ngakala et al., EGUsphere, https://doi.org/10.5194/egusphere-2022-1245, 2022

We thank the anonymous reviewer for this constructive comment to improve our manuscript. Below, we have tried to provide response (blue text) to the reviewer's comment (black text), and we hope that the revised manuscript will satisfy the reviewer.

This version of the paper is improved over the original submission. I have only one minor comment that the authors should address, which is, in line 137, describe in briefly what Holte and Talley method of computing mixed layer depth is rather than require the reader to find the Holte and Talley and read about it there.

Response:

As recommended by the reviewer, we briefly described this method from line 138 in the revised manuscript as follows: "**For individual ocean profiles, this hybrid method models the general shape of each profile, searches for physical features in the profile, and calculates the MLD by using threshold and gradient methods to form a suite of possible MLD 140 values. Then it analyzes the patterns associated with this formed suite in order to select a final MLD estimate.**" (see lines 138-140 "page 5").

---

## Author Response (AR3)

EGUsphere, referee comment RC1
https://doi.org/10.5194/egusphere-2022-1245-RC1, 2022 © Author(s)
2022. This work is distributed under the Creative Commons Attribution
4.0 License.

**Comment on egusphere-2022-1245**

Anonymous Referee #1

Referee comment on "Joint observation-model mixed-layer heat and salt budgets in the eastern tropical Atlantic" by Roy Dorgeless Ngakala et al., EGUsphere, https://doi.org/10.5194/egusphere-2022-1245, 2022

We thank the editor for this comment (black test) and have taken it into account in the final manuscript (see blue text below).

Thank you so much for undertaking the revisions. I am very happy to accept the paper. I note that you have referred in the paper to "saltening" - I am not aware that this is a word so this should be changed to salinification please (perhaps this will be changed by the copy editors anyway). You are right that there ought to be a word saltening to go with freshening, but English is not always logical!

Response:

As recommended by the editor, we have replaced "saltening" by "**salinity increase**" in the final manuscript (see lines 390 "page 17").